# Jasmine: Harnessing Diffusion Prior for Self-Supervised Depth Estimation

**Jiyuan Wang**[1]  **Chunyu Lin**[1†]  **Cheng Guan**[1]  **Lang Nie**[4]  **Jing He**[3]
**Haodong Li**[3]  **Kang Liao**[2]  **Yao Zhao**[1]

[1]**BJTU**  [2]**NTU**  [3]**HKUST**  [4]**CQUPT**  [†]**Corresponding Author**

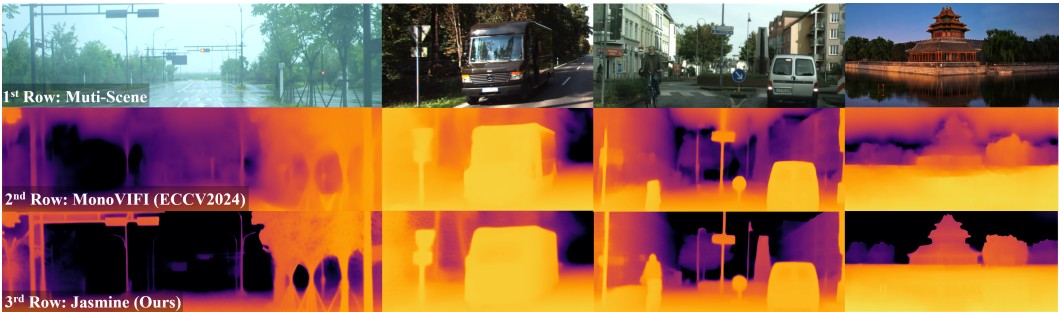

Figure 1: **Without any high-precision depth supervision**, Jasmine achieves remarkably detailed and accurate depth estimation results through zero-shot generalization across diverse scenarios.

## Abstract

In this paper, we propose **Jasmine**, the first Stable Diffusion (SD)-based self-supervised framework for monocular depth estimation, which effectively harnesses SD's visual priors to enhance the sharpness and generalization of unsupervised prediction. Previous SD-based methods are all supervised since adapting diffusion models for dense prediction requires high-precision supervision. In contrast, self-supervised reprojection suffers from inherent challenges (*e.g.*, occlusions, texture-less regions, illumination variance), and the predictions exhibit blurs and artifacts that severely compromise SD's latent priors. To resolve this, we construct a novel surrogate task of mix-batch image reconstruction. Without any additional supervision, it preserves the detail priors of SD models by reconstructing the images themselves while preventing depth estimation from degradation. Furthermore, to address the inherent misalignment between SD's scale and shift invariant estimation and self-supervised scale-invariant depth estimation, we build the Scale-Shift GRU. It not only bridges this distribution gap but also isolates the fine-grained texture of SD output against the interference of reprojection loss. Extensive experiments demonstrate that Jasmine achieves SoTA performance on the KITTI benchmark and exhibits superior zero-shot generalization across multiple datasets. Project page and code are available at here.

## 1  Introduction

Estimating depth from monocular images is a fundamental problem in computer vision, which plays an essential role in various downstream applications such as 3D/4D reconstruction[62, 61], autonomous driving[10], etc. Compared with supervised methods[25, 33, 21], self-supervised monocular depth estimation (SSMDE) mines 3D information solely from video sequences, significantly reducing reliance on expensive ground-truth depth annotations. These methods derive supervision from geometric constraints (*e.g.*, scene depth consistency) through cross-frame reprojection loss, and the ubiquitous video data further suggests an unlimited working potential. However, view reconstruction-based losses suffer from occlusions, texture-less regions, and illumination changes[83], which

39th Conference on Neural Information Processing Systems (NeurIPS 2025).

severely restrict the model's capacity to recover fine-grained details and may cause pathological overfitting to specific datasets.

Recent studies[25] demonstrated that SD possesses powerful visual priors to elevate depth prediction sharpness and generalization, which offers promising potential to address the above limitations. In addition, E2E FT[33] and Lotus[21] further reveal that single-step denoising can achieve better accuracy, which is particularly critical for self-supervised paradigms. It not only accelerates the inference denoising process but also significantly reduces the training costs in self-reprojection supervision, thereby creating opportunities to integrate SD into the SSMDE framework.

However, fine-tuning diffusion models for dense prediction requires high-precision supervision to preserve their inherent priors[25]. Supervised methods typically employ synthetic RGB-D datasets, where clean depth annotations align with the high-quality SD's training data, thereby keeping its latent space intact. In contrast, directly applying self-supervision introduces a critical challenge: reprojection losses or pre-trained depth pseudo-labels propagate perturbed gradients caused by artifacts and blurs into SD's latent space, rapidly corrupting its priors during early training stages. Namely, high-precision "supervision" must exist at the beginning to protect SD's latent space. Such supervision seems impossible in self-supervised learning, but we find a handy and valuable alternative: the RGB image. In fact, the image inherently contains complete visual details, avoids external depth dependencies in self-supervision, and aligns perfectly with SD's original objective of image generation. Therefore, we construct a surrogate task of *mix-batch image reconstruction (MIR)* through a task switcher, where the same SD model alternately reconstructs synthesized/real images and predicts depth maps within each training batch. This strategy repurposes self-supervised reprojection loss to tolerate color variations while maintaining structural consistency[14], intentionally decoupling color fidelity from depth accuracy, and finally preserving SD priors successfully.

Another challenge is the output range of SD's VAE[27] that is inherently bounded within a fixed range, *i.e.*, [-1, 1]. Existing methods typically normalize GT depth maps to this range in the training procedure. During inference, these supervised approaches perform least-squares alignment to recover absolute scale and shift, yielding *scale- and shift-invariant (SSI)* depth predictions. However, self-supervised frameworks rely on coupled depth-pose optimization, which theoretically requires shift invariance to be strictly zero for stable convergence, ultimately producing *scale-invariant (SI)* depth predictions. To bridge this inherent distribution gap, we propose a gated recurrent unit (GRU)-based novel transform module termed *Scale-Shift GRU (SSG)*. It not only iteratively aligns SSI depth to SI depth by refining scale-shift parameters but also acts as a gradient filter, which suppresses anomalous gradients caused by artifact-contaminated in self-supervised training, thereby preserving the fine-grained texture details of SD's output while enforcing geometric consistency.

Extensive experiments show that our proposed method, Jasmine, ❶ achieves the **SoTA performance** among all SSMDEs on the competitive KITTI dataset, ❷ shows **remarkable zero-shot generalization** across multiple datasets (even surpassing models trained with augmented data), ❸ demonstrates **unprecedented detail preservation**. As the first work to bridge self-supervised and zero-shot depth estimation paradigms, we also provide an in-depth analysis of the effects of different depth de-normalization strategies employed in their respective domains. To sum up, the main contributions are summarized as follows:

- We **first** introduce SD into a self-supervised depth estimation framework. Our methods eliminate the dependency on high-precision depth supervision while retaining SD's inherent advantages in detail sharpness and cross-domain generalization.
- We proposed a surrogate task of MIR that anchors SD's priors via self-supervised gradient sharing to avoid SD's latent-space corruption caused by reprojection artifacts.
- We proposed Scale-Shift GRU (SSG) to dynamically align depth scales while filtering noisy gradients to solve SSI versus SI distribution mismatch problems in self-supervised depth estimation.

## 2 Relative Work

**Self-supervised Depth Estimation**    Due to the high costs of GT-depth collection from LiDARs and other sensors, self-supervised depth estimation (SSDE) has gained significant research attention. SSDE can be broadly categorized into stereo-based(learning depth from synchronized image pairs)[52, 14, 12, 3, 2] and monocular-based methods (using sequential video frames)[31, 87, 85, 32, 37, 83, 20, 34, 19, 29]. Additionally, when focusing on inference capability, existing methods can further diverge

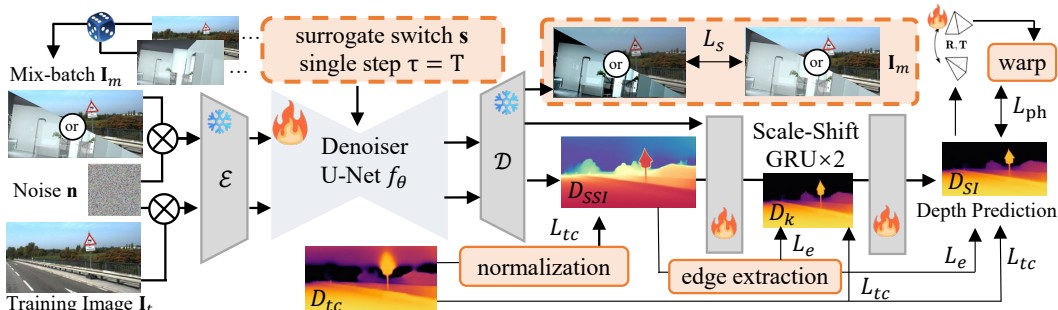

Figure 2: **Finetuning Protocol of Jasmine.** The $\mathbf{I}_t$ and $\mathbf{I}_m$ are each concatenated with $\mathbf{n}$[33] and fed into the VAE encoder $\varepsilon$. Next, the U-Net performs single-step denoising guided by the task switcher $s$, and subsequently decodes the SSI-depth prediction $D_{SSI}$ and the reconstructed image with the $\mathcal{D}$ (Sec. 3.2). Afterward, the $D_{SSI}$ is processed by the SSG for distribution refinement, yielding the final depth estimation $D_{SI}$. The $L_{tc}, L_e, L_{ph}$ and $L_s$ are supervision loss and they are detailed in Sec. 3.4. The edge extraction module is detailed in Sec C

into single-frame and multi-frame approaches[55, 17, 9, 1]. Comparisons between these paradigms are detailed in Sec. E.2. Recently, DepthAnything v1/v2[68, 69] has revealed that we can obtain an accurate single image depth prediction model with strong generalization by training on large-scale image depth pairs. However, we argue that such datasets still remain a small fraction of the ubiquitous video data available. This observation motivates our exploration of the most challenging configuration: training exclusively on video sequences while maintaining single-frame inference capability, thereby laying the groundwork for developing genuinely versatile "DepthAnything" models.

**Diffusion for Depth Perception** As the diffusion paradigm showcases its talents in generative tasks[22, 46, 28, 76, 51, 59], DDP[24] first reformulates depth perception as a depth map denoising task and leads to giant progress. Followers like DDVM[40], MonoDiffusion[44], and D4RD[50] (the latter two are self-supervised methods but employ self-designed diffusion) all demonstrate the advantages of this paradigm in various MDE sub-tasks. Subsequently, the most renowned diffusion model, Stable Diffusion[39], has demonstrated significant potential for depth perception tasks. VPD[84], TAPD[26], and Prior-Diffusion[77] use Stable Diffusion as a multi-modal feature extractor, leveraging textual modality information to improve depth estimation accuracy. Concurrently, Marigold[25] and GeoWizard[11] enhanced model generalization and detail preservation by fine-tuning Stable Diffusion, capitalizing on its prior training with large-scale, high-quality datasets. Afterward, E2E FT[33] and Lotus[21] further accelerated inference by optimizing the noise scheduling process. In this work, our Jasmine continues these works and extends SD to the field of self-supervision.

## 3 Methods

In this section, we will introduce the foundational knowledge of the SSMDE and SD-based MDE (Sec. 3.1), the surrogate task of mix-batch image reconstruction (MIR, Sec. 3.2), the scale-shift adaptation with GRU (Sec. 3.3) and the SD finetune protocol specified for self-supervision (SSG, Sec. 3.4). An overview of the whole framework is shown in Fig. 2 and its training pseudocode is shown in Algorithm 1.

### 3.1 Preliminaries

**Self-Supervised Monocular Depth Estimation** makes use of the adjacent frames $\mathbf{I}_{t'}$ to supervise the output depth with geometric constraints. Given the current frame $\mathbf{I}_t$, the MDE model as $\mathcal{F}: \mathbf{I}_t \to D \in \mathbb{R}^{W \times H}$, we can synthesize a warped current frame $\mathbf{I}_{t' \to t}$ with:

$$\mathbf{I}_{t' \to t} = \mathbf{I}_{t'} \langle \text{proj}(D, \mathcal{T}_{t \to t'}, K) \rangle, t' \in (t-1, t+1), \tag{1}$$

where $\mathcal{T}_{t \to t'}$ denotes the relative camera poses obtained from the pose network, $K$ denotes the camera intrinsics, and $\langle \cdot \rangle$ denotes the grid sample process. Then, we can compute the photometric reconstruction loss between $\mathbf{I}_t$ and $\mathbf{I}_{t' \to t}$ to constrain the depth:

$$L_{ph}(\mathbf{I}_t, \mathbf{I}_{t' \to t}) = \eta_{p1}(1 - \text{SSIM}(\mathbf{I}_t, \mathbf{I}_{t' \to t}))/2 + \eta_{p2} \|\mathbf{I}_t - \mathbf{I}_{t' \to t}\|. \tag{2}$$

**Stable Diffusion-based Monocular Depth Estimation** reformulates depth prediction as an image-conditioned annotation generation task. Typically, given the image $\mathbf{I}$ and processed GT depth $\mathbf{y}$, SD first encodes them to the low-dimension latent space through a VAE encoder $\varepsilon$, as $(\mathbf{z}^{\mathbf{I}}, \mathbf{z}^{\mathbf{y}}) = \varepsilon(\mathbf{I}, \mathbf{y})$.

Afterward, gaussian noise is gradually added at levels $\tau \in [1, T]$ into $\mathbf{z^y}$ to obtain the noisy sample, with $\mathbf{z_\tau^y} = \sqrt{\overline{\alpha}_\tau}\mathbf{z^y} + \sqrt{1 - \overline{\alpha}_\tau}\epsilon$, then the model learns to iteratively reverse it by removing the predicted noise:

$$\hat{\epsilon} = f_\theta^\epsilon(\mathbf{z_{\tau-1}^y}|\mathbf{z_\tau^y}, \mathbf{z^I}), \tag{3}$$

and finally decodes the depth prediction with $D = \mathcal{D}(\mathbf{z_0^y})$. Here $\epsilon \sim \mathcal{N}(0, I)$, $f_\theta^\epsilon$ is the $\epsilon$-prediction U-Net, $\overline{\alpha}_t := \prod_{s=1}^{t}(1 - \beta_s)$, $\mathcal{D}$ is the VAE decoder and $\{\beta_1, \beta_2, \ldots, \beta_T\}$ is the noise schedule with $T$ steps.

Equation 1 demonstrates that the self-supervised approach needs the depth prediction $D$ for image warping. However, obtaining $\mathbf{z_0^y}$ requires iterative computation of Eq. 3, which becomes computationally infeasible considering the enormous size of SD models. Fortunately, Lotus and E2E FT[33, 21] demonstrate that we can obtain comparable results with single-step denoising and directly predict depth, $D = \mathcal{D}\left(f_\theta^z\left(\mathbf{z_\tau^y}, \mathbf{z^I}\right)\right), \tau = T$, which makes it possible to train SD with self-supervision. The step-by-step workflow is detailed in Sec B.

### 3.2 Surrogate Task: Image Reconstruction

**Self-supervision Compromise SD Prior.** The photometric reconstruction losses (Eq. 2) inevitably introduce the supervision with noise and artifacts due to occlusions, texture-less regions, and photometric inconsistencies. As shown in Fig. 3 (a), consider a scenario where relative camera poses $\mathcal{T}_{t \to t'}$ represents a pure horizontal translation. The point $p$—which should ideally have 5 pixels of disparity (reciprocal of depth)—becomes occluded. Instead, it must displace 10 to compensate for incorrect pixel matches, resulting in erroneous depth alignment with point $q$ (a detailed explanation is provided in Sec. F). This phenomenon propagates to neighboring points, collectively eroding structural details (*e.g.*, the tree's edge) while generating imprecise supervisory signals that rapidly degrade SD's fine-grained prior knowledge.

To preserve these details, we notice that both SSMDE and SD inherently rely on image consistency: SSMDE uses photometric constraints (Eq. 2), while SD's training directly minimizes image generation errors. Inspired by this, we propose a surrogate task: image reconstruction. Concretely, following [11], we design a switcher $s \in \{s_x, s_y\}$ to alternate the U-Net $f_\theta^z$ between the main and surrogate tasks. When activated by $s_x$, we have $D = \mathcal{D}\left(f_\theta^z\left(s_x, \mathbf{z_\tau^y}, \mathbf{z^I}\right)\right)$. In contrast, we have $\mathbf{I} = \mathcal{D}\left(f_\theta^z\left(s_y, \mathbf{z_\tau^y}, \mathbf{z^I}\right)\right)$. Notably, the switcher $s$ is a processed one-hot vector and it is combined with the time embeddings fed into the $f_\theta^z$. This allows the U-Net to condition its internal operations on the currently selected task. Therefore, we follow the SD paradigm and initially formulate the surrogate loss as:

$$L_s = ||\mathbf{z^I} - f_\theta^z(s_y, \mathbf{z_\tau^y}, \mathbf{z^I})||^2. \tag{4}$$

**Mix-batch Images Reconstruction with Photometric Supervision.** We show that it is possible to preserve the SD priors by introducing a compact surrogate task. However, our experiments reveal that naively applying Eq. 4 for SSMDE optimization yields suboptimal results (Fig. 3(c)). Through empirical investigation, we identify three critical insights: **1)** Inferior reconstructed images (*e.g.*, KITTI) introduce block artifacts (Fig. 3(c)). We attribute this to the latent space operating at $\frac{1}{8}$ resolution: When inputs align with the pre-trained $\varepsilon$ and $\mathcal{D}$, smooth supervision is achieved. Conversely, each latent pixel supervision manifests as 8×8 block artifacts in prediction. **2)** Introducing high-quality synthesized images (maintaining self-supervision compliance) offers a potential solution, but exclusive training on these data causes the model to only excel at reconstructing synthetic images but fails to generalize this capability to reduce depth estimation blurriness(Fig. 3 (d)). **3)** Mixing these images within a training batch allows synthesized images to anchor the model to latent priors, while real-world data enforces geometric structure alignment, is a possible solution. But this strategy shows notable sensitivity to mix rate $\lambda$ (Fig. 3 (g), (e) is a failure scene).

To address the mismatch between VAE and image, we proposed to replace Eq. 4 with the photometric loss $L_{ph}$ (Eq. 2) in the image domain. Compared to Eq. 4, $L_{ph}$ emphasizes structural consistency rather than color fidelity, which aligns better with depth estimation objectives. As shown in Fig. 3(f, g), this supervision not only makes MIR robust to $\lambda$ but also significantly improves the estimation quality. Therefore, we take the Hypersim dataset(4.2.1), a photorealistic synthetic dataset specifically designed for geometric learning, as the auxiliary image and update Eq. 4 with

$$\mathbf{I_h} = \phi(\mathbf{I}_K, \mathbf{I}_H, \lambda), L_s = L_{ph}(\mathbf{I_h}, \mathcal{D}(f_\theta^z(s_y, \mathbf{z_\tau^y}, \mathbf{z^I}))), \tag{5}$$

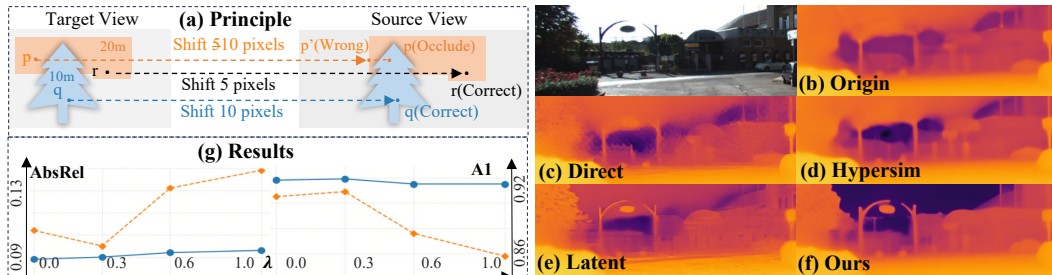

Figure 3: **The attempts to preserve the SD prior.** The meanings of (a)-(f) are detailed in Sec. 3.2. Notably, while (e) demonstrates superior visual quality, it erroneously interprets surface textures (*e.g.*, house windows) as depth edges. (g) shows the performance variations under different $\lambda$ settings for photometric supervision (Eq. 5) and latent supervision (Eq. 4). The complete metrics and their definitions are provided in Sec. E.4.

where $\phi$ denotes random choice; $\mathbf{I}_K$ and $\mathbf{I}_H$ are KITTI and Hypersim images, respectively.

**In summary**, MIR constructs each training batch by randomly selecting images from these two datasets, and supervises the reconstructed image with the photometric loss in Eq. 5.

**Analysis of Auxiliary Images** The specific synthetic data usage may raise concerns about applicability boundaries. To clarify, we conduct additional experiments and present three insights about the relationship between auxiliary data and performance: (1) Synthetic images are not essential; our surrogate task maintains efficacy with real-world imagery. (2) The dataset scale proves non-critical, as competitive performance emerges with samples under 1k. (3) Domain divergence between auxiliary and primary datasets enhances results and is even more important than image quality. Please refer to Sec. 4.4 for detailed experimental support.

This analysis and related experiments reveal MIR is a highly promising training paradigm. It not only imposes no inherent limitations on any dense prediction tasks but also challenges the notion that fine-tuning SD requires high-quality annotation. Even with legacy datasets like KITTI, we can still leverage readily available images to effectively utilize SD priors and enhance depth estimation sharpness.

### 3.3 Scale-Shift GRU

**The misalignment of SSI-SI depth.** We first analyze the training procedure of SSMDE. Denoting the relative camera poses $\mathcal{T}_{t \to t'}$ as $[\mathbf{R}|\mathbf{T}]$, we can further expand the *proj* process in Eq. 1 with:

$$\zeta' D' = K(\mathbf{R}K^{-1}\zeta D + \mathbf{T}), \tag{6}$$

where $\zeta, \zeta'$ denote homogeneous coordinates and $D, D'$ represent depths in $\mathbf{I}_t$ and $\mathbf{I}_{t'}$, respectively. Afterward, the coordinate $\zeta'[u,v]$ in $\mathbf{I}_{t'}$ maps to $[u,v]$ in $\mathbf{I}_{t' \to t}$ and we can grid sample every pixel to get $\mathbf{I}_{t' \to t}$. However, this mapping is not unique. We can scale both sides of the equation with $s_c$:

$$\zeta' s_c D' = K(\mathbf{R}K^{-1}\zeta s_c D + s_c \mathbf{T}), \tag{7}$$

where both $\mathbf{T}$ and $s_c\mathbf{T}$ represent valid relative poses. But if we further introduce a shift $s_h$, we can derive the formula (detailed derivations at Sec. A) to obtain:

$$K^{-1}\zeta' g_1(D') = \mathbf{R}K^{-1}\zeta s_h + g_2(\mathbf{T}), \tag{8}$$

where $g_1(\cdot)$ and $g_2(\cdot)$ are affine transformations (SSI depth is affine depth). The above equation means that the affine depth of any scene ($g_1(D')$) can appear as a plane from a certain perspective. This plane has the depth $s_h$ and the extrinsic transformation of this perspective and the original one is $[\mathbf{R}|\mathbf{T}]$, which is undoubtedly impossible (more explanations in Sec. A). Thus, the shift $s_h$ does not exist under the geometric constraints, and SSMDE predicts Scale-Invariant Depth $D_{SI}$.

Additionally, we analyze the training process of SD-based MDE. The processed GT depth $\mathbf{y}$ mentioned in Sec. 3.1 satisfies the VAE's[27] inherent boundary [-1, 1], typically obtained by:

$$\mathbf{y} = ((D_{GT} - D_{GT,2})/(D_{GT,98} - D_{GT,2}) - 0.5) \times 2, \tag{9}$$

where $D_{GT,i}$ corresponds to the $i\%$ percentiles of individual depth maps. Obviously, compared to the raw depth, $\mathbf{y}$ is normalized and differs from $D_{GT}$ by an absolute scale and shift, which can

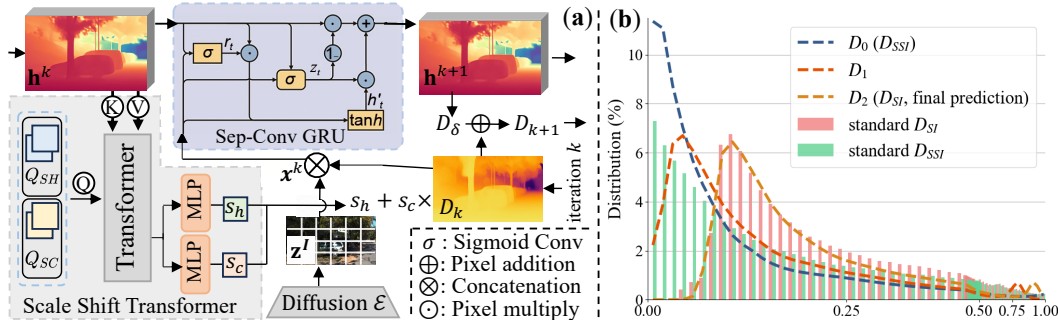

Figure 4: **(a): Model Structure of SSG**. It corresponds to the gray rectangle shown in Fig. 2, standing for an iteration within two consecutive ones. The pipeline of SSG is comprehensively described in Sec. 3.3 (DepthHead is omit in (a) for clear). **(b): Depth distribution alignment visualization**. We statistically analyze each stage of Jasmine's SSG module on the KITTI test set. The standard SI and SSI depths are obtained by applying Eq. 9 and dividing the maximum value to the depth GT, respectively.

be recovered by the least squares alignment in evaluation. Consequently, SD-based MDE predicts Scale-Shift-Invariant Depth $D_{SSI}$.

This inherent distribution gap between SSI and SI depth creates barriers for SD integration, and the SSG is specifically designed to fix it.

**The Design of SSG.** Transforming the depth distribution from SSI depth to SI depth requires profound scene understanding, and the scale ($s_c$) and shift ($s_h$) factors are tightly coupled. Therefore, as shown in Fig. 4(a), compared to traditional GRU, SSG introduces a core component Scale-Shift Transformer (SST), and modifies the iterative prediction formula:

$$From \quad D_{k+1} = D_\delta + D_k \quad To \quad D_{k+1} = D_\delta + s_c \cdot D_k + s_h, \tag{10}$$

where $D_\delta = \text{DepthHead}(\mathbf{h}^{k+1})$, $k$ denotes the iteration step and $\mathbf{h}$ denotes the hidden state (preliminaries of GRU in Sec D). Specifically, the SST employs learnable scale/shift queries ($Q_{SC}/Q_{SH}$) that interact with SD's hidden states (keys/values) via cross-attention. The output vector is subsequently split and processed by MLPs to produce $s_c$ and $s_h$. For the hidden state update, to enhance spatial awareness, the current input $\mathbf{x}^k$ is defined as the concatenation of image features $\mathbf{z}^I$ and the current depth $D_k$. The hidden state $\mathbf{h}^k$ evolves via a standard GRU iteration to produce the refined hidden state $\mathbf{h}^{k+1}$ and subsequently update $D_k$ to $D_{k+1}$.

To balance computational efficiency and GRU's iterative benefits, we employ two GRU iterations: starting from the initial depth $D_0$ ($D_{SSI}$) to sequentially produce $D_1$ and $D_2$ ($D_{SI}$). As shown in Fig. 4(b), the distribution of $D_0$ tends to align with the standard SSI depth distribution, while $D_1$ and $D_2$ progressively converge towards the SI depth. This clearly demonstrates that SSG iteratively aligns SSI depth to SI depth by refining the scale-shift parameters. GRU is preferred over other architectures due to its reset gate mechanism. During training, the reset gate $\mathbf{r}$ can prevent the backpropagation of anomalous gradients to the former step by selectively resetting parts of the hidden state. Therefore, this mechanism enables the fine-grained $D_{SSI}$ to filter out erroneous supervision signals from reprojection losses and exhibit richer details than $D_{SI}$ (shown in Fig. 2). To preserve these fine-grained details in the final prediction, we further constrain the edge alignment between $D_{SSI}$ and $D_{SI}$ with an edge extraction module, detailed in Sec. C.

### 3.4 Steady SD Finetune with Self-Supervision

As the **first** framework to finetune SD with self-supervision, we encountered a novel challenge: training instability, which is mainly due to the SD's enormous size, joint training across modules, and indirect self-supervisory mechanisms. To enhance the convergence reliability and reproducibility, we explored a straightforward approach by introducing a pre-trained self-supervised teacher model (*e.g.*, MonoViT) to estimate $D_{tc}$ as pseudo labels. $D_{tc}$ provides direct supervision but has a performance upper bound, which can stabilize model training in the early stages while gradually decreasing loss weights throughout the training process:

$$L_{tc} = (L_B (\text{norm}(D_{tc}), D_{SSI}) + L_B (D_{tc}, D_1) \cdot \eta_{t1} + L_B (\text{filter}(D_{tc}), D_{SI}) \cdot \eta_{t2}) / \eta_{\text{step}}, \tag{11}$$

where $L_B$ is the Berhu Loss, "norm" is the [-1,1] normalization, and "filter" are adaptive strategies to avoid performance bottlenecks. Through extensive experimentation and error bar analysis, we

demonstrate that this pseudo-label training proves particularly crucial for steady training in complex, multi-module self-supervised systems. The implementation details are in Sec. C. Finally, the total training loss of the Jasmine model is:

$$L = L_s + L_{ph} + L_{tc} + L_a \cdot \eta_a, \tag{12}$$

where $L_s$ refers to Eq.5, $L_{ph}$ refers to Eq. 2, $L_{tc}$ refers to Eq. 11, and $L_a$ is some auxiliary adjustment losses (*e.g.,* gds loss[34] $L_{GDS}$, edge loss $L_e$, etc.) with a tiny weight. They will be detailed in supplementary material Sec. C.

## 4 Experiment

### 4.1 Implement Details

We implement the proposed Jasmine using Accelerate[16] and PyTorch[35] with Stable Diffusion v2[39] as the backbone. Following the pipeline in Fig. 2, we disable text conditioning while maintaining most hyperparameter consistency with E2E FT[33]. The loss weights specified in Sec. 3 are empirically configured as:

$$\eta_a = 8e-3, \eta_{t1} = 0.6, \eta_{t2} = 0.9, \eta_{p1} = 0.85, \eta_{p2} = 0.15, \eta_{\text{step}} = \max(1, 30 \cdot (\text{step}_{now}/\text{step}_{max})).$$

Training uses the AdamW optimizer[30] with a base learning rate of $3e-5$. All experiments are conducted on 8 NVIDIA A800 GPUs with a total batch size of 32, training for a total of 25k training steps, requiring around 1 day. Following [15], we also employed standard data augmentation techniques (horizontal flips, random brightness, contrast, saturation, and hue jitter).

### 4.2 Evaluation

#### 4.2.1 Datasets

Unless specified, all datasets are finally resized to 1024×320 resolution for training.

**Training Datasets**. KITTI[13]: Following the previous work[15], we mainly conduct our experiments on the widely used KITTI dataset. We employ Zhou's split[86] containing 39,810 training and 4,424 validation samples after removing static frames. The evaluation uses 697 Eigen raw test images with metrics from [15], applying 80m ground truth clipping and Eigen crop preprocessing[8]. Hypersim[38]: This photorealistic synthetic dataset (461 indoor scenes) contributes approximately 28k samples from its official training split for mix-batch image reconstruction. Each iteration uses random crops from the original 1024×768 to 1024×320 resolution.
**Zero-shot Evaluation Datasets** DrivingStereo[67]: Contains 500 images per weather condition (fog, cloudy, rainy, sunny) for zero-shot testing. CityScape[7]: Evaluated on 1,525 test images with dynamic vehicle-rich urban scenes, using ground truth from [55].
**MIR Analysis Datasets** ETH3D[41]: We resize this high-resolution (6048×4032) dataset to 4K resolution, then randomly cropped to 1024×320 per iteration (898 total samples). Virtual KITTI[4] is a synthetic street scene dataset. We processed this dataset identically to real KITTI data.

#### 4.2.2 Performance Comparison

For all the evaluations, only Jasmine*, Marigold, E2E FT, and Lotus adopt the least squares alignment. The other self-supervised methods use median alignment. The definitions and differences of these alignments are detailed discussed in Sec. 4.3. The meaning of each metric is detailed in Sec. G.

**KITTI result** To fully demonstrate the advantages of our approach, we compare Jasmine against the most efficient SSMDE models and SoTA SD-based methods. As shown in Table 1, *Jasmine achieves the best performance* across all metrics on the KITTI benchmark. Notably, our method makes significant progress on the $a_1$ metric, reflecting an overall improvement in depth estimation accuracy. This systematic advancement stems from the rich prior knowledge of SD. As shown in Fig. 1, without any specialized design for reflective surfaces, our method can accurately distinguish scene elements from their reflections. Additionally, the first row of Fig. 5 also demonstrates that our approach *preserves structural details better* than existing methods. Moreover, when compared to the other SD-based zero-shot models, none of which, including Jasmine*, use the ground truth depth of KITTI, our method demonstrates a substantial performance advantage, further highlighting the value of self-supervised techniques.

Table 1: **Quantitative results on the KITTI dataset.** For the `error-based metrics`, the lower value is better; and for the `accuracy-based metrics`, the higher value is better. The best and second-best results are marked in **bold** and underline. Jaeho et al* is a combined model that applies both Jaeho's[34] (handle dynamic objects) and TriDepth's[5] (solve edge flatten) approach. Jasmine* and Jasmine are the same model but use different alignments (discussed in Sec. 4.3). In the data column, Syn, K, and H represent the synthetic, KITTI, and Hypersim datasets, respectively. The number of images and depth labels usage are in brackets. All experiments are conducted at 1024×320 resolution (Performance of Marigold/E2E FT/Lotus is robust to this resolution).

| Method | Venue | Notes | Data | AbsRel | SqRel | RMSE | RMSElog | $a_1$ | $a_2$ | $a_3$ |
|---|---|---|---|---|---|---|---|---|---|---|
| Marigold[25] | CVPR2024 | ZeroShot | Syn(74K+74K) | 0.120 | 0.672 | 4.033 | 0.184 | 0.874 | 0.968 | 0.985 |
| E2E FT[33] | WACV2025 | ZeroShot | Syn(74K+74K) | 0.112 | 0.649 | 4.099 | 0.180 | 0.890 | 0.969 | 0.985 |
| Lotus[21] | ICLR2025 | ZeroShot | Syn(59K+59K) | 0.110 | 0.611 | 3.807 | 0.175 | 0.892 | 0.970 | 0.986 |
| Jasmine* | - | Mono | KH(68K+0) | **0.102** | **0.540** | **3.728** | **0.162** | **0.907** | **0.973** | **0.987** |
| Monodepth2[15] | ICCV2019 | Mono | K(40K+0) | 0.115 | 0.882 | 4.701 | 0.190 | 0.879 | 0.961 | 0.982 |
| HR-Depth[31] | AAAI2021 | Mono | K(40K+0) | 0.106 | 0.755 | 4.472 | 0.181 | 0.892 | 0.966 | 0.984 |
| R-MSFM6[87] | ICCV2021 | Mono | K(40K+0) | 0.108 | 0.748 | 4.470 | 0.185 | 0.889 | 0.963 | 0.982 |
| DevNet[85] | ECCV2022 | Mono | K(40K+0) | 0.100 | 0.699 | 4.412 | 0.174 | 0.893 | 0.966 | 0.985 |
| DepthSegNet[32] | ECCV2022 | Mono | K(40K+0) | 0.099 | 0.624 | 4.165 | 0.171 | 0.902 | 0.969 | 0.985 |
| SD-SSMDE[37] | CVPR2022 | Mono | K(40K+0) | 0.098 | 0.674 | 4.187 | 0.170 | 0.902 | 0.968 | 0.985 |
| MonoViT[83] | 3DV 2022 | Mono | K(40K+0) | 0.096 | 0.714 | 4.292 | 0.172 | 0.908 | 0.968 | 0.984 |
| LiteMono[81] | CVPR2023 | Mono | K(40K+0) | 0.102 | 0.746 | 4.444 | 0.179 | 0.896 | 0.965 | 0.983 |
| DaCCN[20] | ICCV2023 | Mono | K(40K+0) | 0.094 | 0.624 | 4.145 | 0.169 | 0.909 | 0.970 | 0.985 |
| Jaeho et al*.[5, 34] | CVPR2024 | Mono | K(40K+0) | 0.091 | 0.604 | 4.066 | 0.164 | 0.913 | 0.970 | **0.986** |
| RPrDepth[19] | ECCV2024 | Mono | K(40K+0) | 0.091 | 0.612 | 4.098 | 0.162 | 0.910 | 0.971 | **0.986** |
| Mono-ViFI[29] | ECCV2024 | Mono | K(40K+0) | 0.093 | 0.589 | 4.072 | 0.168 | 0.909 | 0.969 | 0.985 |
| Jasmine | - | Mono | KH(68K+0) | **0.090** | 0.581 | **3.944** | **0.161** | **0.919** | **0.972** | **0.986** |

Table 2: **Quantitative zero-shot results on the CityScape and DrivingStereo dataset and its variants (Rainy, Cloudy, Foggy).** Alignment protocols and annotation rules follow Table 1's specifications. AbsRel, RMSE, and $a_1$ metrics are shown.

| Method | DrivingStereo | | | Rainy | | | CityScape | | | Cloudy | | | Foggy | | |
|---|---|---|---|---|---|---|---|---|---|---|---|---|---|---|---|
| | AbsRel | RMSE | $a_1$ | AbsRel | RMSE | $a_1$ | AbsRel | RMSE | $a_1$ | AbsRel | RMSE | $a_1$ | AbsRel | RMSE | $a_1$ |
| Marigold[25] | 0.178 | 6.638 | 0.749 | **0.148** | 6.770 | 0.801 | 0.164 | 6.632 | 0.763 | 0.173 | 6.881 | 0.751 | 0.146 | 6.545 | 0.798 |
| E2E FT[33] | 0.160 | 5.437 | 0.795 | 0.164 | 6.671 | 0.793 | 0.160 | 6.944 | 0.792 | 0.157 | 5.522 | 0.797 | 0.141 | 6.034 | 0.836 |
| Lotus[21] | 0.173 | 5.816 | 0.771 | 0.167 | 6.675 | 0.775 | 0.147 | 6.582 | 0.824 | 0.159 | 5.640 | 0.795 | 0.150 | 6.173 | 0.798 |
| Jasmine* | **0.134** | **4.666** | **0.854** | 0.159 | 6.071 | 0.825 | **0.107** | 5.000 | 0.907 | 0.134 | 4.762 | 0.846 | 0.113 | 4.883 | 0.897 |
| MonoDepth2[15] | 0.191 | 8.359 | 0.770 | 0.260 | 12.577 | 0.609 | 0.158 | 8.185 | 0.783 | 0.192 | 10.07 | 0.775 | 0.156 | 10.425 | 0.799 |
| MonoViT[83] | 0.150 | 7.657 | 0.815 | 0.190 | 9.407 | 0.724 | 0.140 | 7.913 | 0.802 | 0.134 | 7.280 | **0.849** | 0.107 | 7.899 | 0.882 |
| Mono-ViFI[29] | 0.158 | 6.723 | 0.798 | 0.400 | 13.960 | 0.484 | 0.134 | 7.372 | 0.817 | 0.154 | 6.883 | 0.800 | 0.160 | 8.494 | 0.769 |
| WeatherDepth[49] | 0.166 | 6.986 | 0.796 | 0.166 | 8.844 | 0.748 | 0.137 | 6.515 | 0.837 | 0.167 | 7.566 | 0.793 | 0.132 | 7.679 | 0.859 |
| Jasmine | **0.136** | 5.340 | 0.850 | 0.160 | 7.194 | 0.787 | 0.123 | 6.618 | 0.852 | 0.133 | 5.651 | 0.849 | **0.098** | 5.702 | **0.902** |

**Generalization Results** Compared to in-domain evaluation, Jasmine exhibits even more remarkable results in zero-shot generalization. Due to the large number of comparison datasets and the unavailability of some models' open-source code, we only compare with Monodepth2 (the most classic model), MonoViT (renowned for its robustness), MonoViFi (the latest model), and WeatherDepth (trained with additional weather-augmented data, totaling 278k samples), all of which are self-supervised models. In fact, MonoViT already surpasses the zero-shot capability for nearly all models in the SSMDE, making it a sufficiently strong baseline[42]. Additionally, SD-based methods have excelled in generalization, allowing us to conduct a fair zero-shot comparison here. As shown in Table 2, Jasmine demonstrated state-of-the-art performance on the datasets of Cityscape

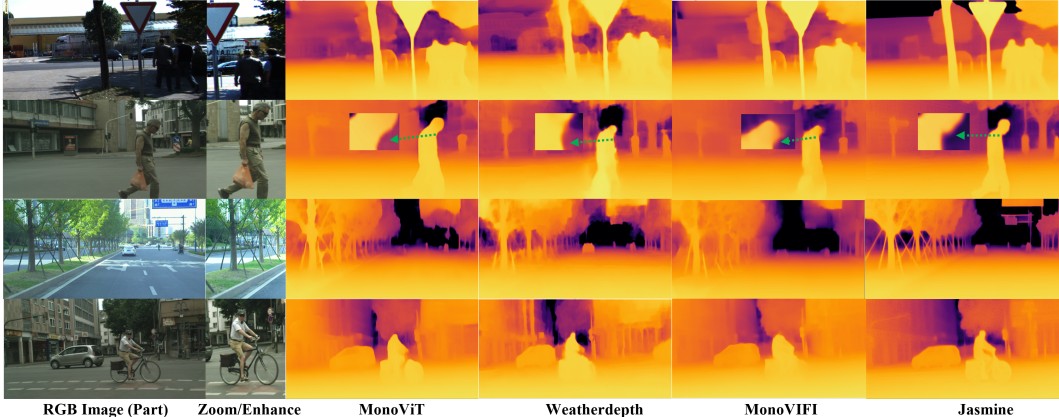

| RGB Image (Part) | Zoom/Enhance | MonoViT | Weatherdepth | MonoVIFI | Jasmine |

Figure 5: **Qualitative results** on KITTI, DrivingStereo, and CityScape datasets. We compare Jasmine with the most generalizable and best-performing SSMDE methods in both in-domain and zero-shot scenarios.

and four weather scenarios of driving stereo. Remarkably, our method *maintains effectiveness in out-of-distribution (OOD) rainy conditions even without specialized training on weather-enhanced datasets* like WeatherKITTI[49]. As illustrated in Fig. 1, Jasmine successfully identifies water surface reflections while producing refined depth estimates. The fine-grain estimation details are further demonstrated in Fig. 5, pedestrian chins (second row), tree support structures (third row), and bicycle-rider contours (fourth row) are all predicted delicately and precisely. These sharp results were completely disrupted by the reprojection loss in previous self-supervised methods.

**Further Analysis** We also deeply compare Jasmine with other SSDE configurations (stereo training and multi-frame inference) and on KITTI improved GT benchmark in Sec. E.2, E.1.

### 4.3 Analysis of Depth De-normalization

The performance gap between Jasmine and Jasmine* in Tables 1 and 2 highlights the impact of de-normalization strategies—median alignment for self-supervised methods and LSQ alignment for zero-shot settings. As the first work bridging these two domains, we provide an in-depth analysis of how these choices affect evaluation.

Depth de-normalization refers to the process of transforming the model's predicted depth values back to the GT distribution for metric evaluation. The typical de-normalization strategies include:

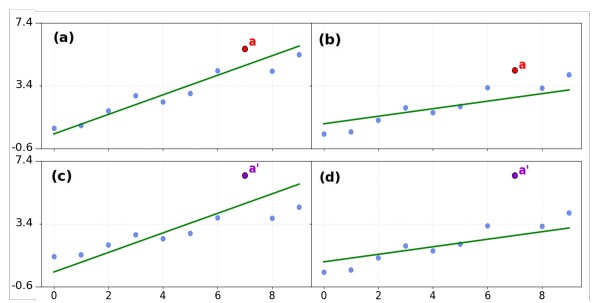

Figure 6: **Comparison of different de-normalization schemes.** The blue points are predictions after alignment and the green line is the ideal GT depth. Sub-figures (a,c) are the results of LSQ alignment, while (b,d) are median alignment.

❶ No operation: Models predict metric depth [23]: $D_{eval} = D_{pred}$.
❷ Median Alignment: Models predict SI Depth [15], which can be recovered by scaling the ratio of the median of the GT depth to the median of the predicted depth: $D_{eval} = D_{pred} \cdot \text{median}(D_{GT})/\text{median}(D_{pred})$.
❸ Least-Squares (LSQ) Alignment: Models predict SSI Depth [25], which can be recovered by affine transformation (scaled and shifted) to best fit the GT depth. The evaluated depth is given by: $D_{eval} = s^* D_{pred} + t^*$. where the optimal scale $s^*$ and shift $t^*$ are determined by minimizing the sum of squared differences: $(s^*, t^*) = \underset{s,t}{\arg\min} \sum_i ((s \cdot D_{pred,i} + t) - D_{GT,i})^2$.

From Tables 1 and 2, we have following observations:
**Metric inconsistency:** For the exact same model, the evaluation metrics can vary dramatically depending on the de-normalization method, making direct comparison unfair.
**Metrics characteristics:** Under LSQ alignment, quadratic metrics (e.g., SqRel, RMSE) are usually better, but first-order metrics (e.g., AbsRel) and overall accuracy ($a_1$) are often worse.
**Scenario suitability:** For in-domain training, median alignment is generally superior, while in zero-shot scenarios, LSQ alignment is usually stronger.

As shown in Fig. 6 (a, c), LSQ alignment tends to accommodate outliers $a$ and $a'$, resulting in a larger overall shift in the alignment. In contrast, in sub-figures (b, d), outliers have little effect on the median, so the accuracy for the majority of points is preserved. In the context of depth estimation, $a$ and $a'$ can represent the model's predictions for regions that are difficult to estimate. Clearly, for quadratic metrics, under median alignment, the large error between $a'$ and the GT in (d) will be further amplified by squaring, leading to a drop in the overall metric. However, when computing overall accuracy, $a'$ or $a$ are typically outside the threshold for $a_1$ and thus do not affect it, while the originally accurate predictions for other points become less accurate due to the shift introduced by LSQ alignment. Similarly, first-order metrics also degrade due to these shifts. This explains why Jasmine* achieves better RMSE and SqRel but worse $a_1$ and AbsRel compared to Jasmine.

For the last observation, in in-domain scenarios (Table 1), Jasmine outperforms Jasmine* because the shift-free estimation learned by Jasmine is disrupted by LSQ alignment, introducing a suboptimal shift and degrading performance. In out-of-domain scenarios (Table 2), Jasmine* performs better, as the depth distributions of different datasets may differ significantly, and using least-squares to estimate the best fit is clearly a better choice.

## 4.4 Ablation Study

As shown in Table 3, we conduct ablation studies to validate our designs. Firstly, in sub-table **(a)**, we gradually tested the effects of our basic components, such as SD prior, MIR, and SSG. The SD prior proves most critical - training from scratch (ID0 *vs.* ID1) causes catastrophic failure (AbsRel↑473%, RMSE↑206%). The other experiments also demonstrate that disabling MIR (ID4) or SSG (ID3) degrades performance by 47%/43% in AbsRel, proving the necessity of depth distribution alignment and SD detail preservation.

In sub-table **(b)**, we further melted down our proposed SSG. ID (6) reveals that the naive GRU can initially solve the distribution misalignment by estimating the $D_\delta$ (Eq. 4). However, the scale difference between SSI and SI depth makes it difficult to restore through linear addition. Therefore, after introducing SST, the overall model performance is further enhanced by 10%, ultimately achieving SoTA performance. A comprehensive analysis of MIR was conducted in sub-table **(c)** and we can draw similar conclusions to Sec. 3.2. Jasmine significantly outperforms the alternatives, such as KITTI/synthetic-only reconstruction (IDs (8,9) and Fig. 3 (c,d)) and latent-space supervision (ID (10) and Fig. 3 (e)), which strongly proves the effectiveness of our proposed MIR. The analysis of auxiliary images is presented in sub-table **(d)**. The experiments in IDs (0, 13, 14) indicate that, compared to real/synthesis datasets, the content of the dataset is more important, and diverse scenes offer greater benefits than street views (virtual KITTI images) similar to our primary dataset, KITTI. Furthermore,

Table 3: **Ablation Studies.** vK and Hy mean the virtual KITTI and Hypersim datasets. Dataset/$n$ denote we downsample the image to $\frac{1}{n}$ resloution(*i.e.* Hy/1.6 means downsample to 640×192, where 640=1024/1.6) and resize them back.

| (ID) Method | AbsRel | SqRel | RMSE | $a_1$ | $a_2$ |
|---|---|---|---|---|---|
| Ours | | | | | |
| (0) Jasmine | 0.090 | 0.581 | 3.944 | 0.919 | 0.972 |
| (a) Basic Component | | | | | |
| (1) w/o SD Prior | 0.516 | 6.019 | 12.06 | 0.258 | 0.501 |
| (2) w/o MIR+SSG | 0.175 | 2.264 | 7.969 | 0.790 | 0.929 |
| (3) w/o SSG | 0.129 | 0.938 | 4.470 | 0.872 | 0.956 |
| (4) w/o MIR | 0.132 | 0.673 | 4.271 | 0.852 | 0.967 |
| (b) Scale-Shift GRU | | | | | |
| (5) w/o SSG | 0.129 | 0.938 | 4.470 | 0.872 | 0.956 |
| (6) w/o SST | 0.098 | 0.715 | 4.350 | 0.909 | 0.969 |
| (c) Mix-batch Image Reconstruction | | | | | |
| (7) w/o MIR | 0.132 | 0.673 | 4.271 | 0.852 | 0.967 |
| (8) Direct | 0.129 | 0.679 | 4.385 | 0.858 | 0.962 |
| (9) Only Hy | 0.106 | 0.614 | 4.181 | 0.901 | 0.970 |
| (10) Latent space | 0.095 | 0.606 | 4.138 | 0.909 | 0.970 |
| (d) Auxiliary Image Analysis | | | | | |
| (12) KITTI | 0.095 | 0.616 | 4.040 | 0.912 | 0.972 |
| (13) KITTI+ETH3D | 0.090 | 0.586 | 3.937 | 0.916 | 0.972 |
| (14) KITTI+vK | 0.094 | 0.606 | 4.068 | 0.911 | 0.972 |
| (15) KITTI+Hy/4 | 0.091 | 0.596 | 3.943 | 0.917 | 0.972 |
| (16) KITTI+Hy/1.6 | 0.090 | 0.591 | 3.971 | 0.918 | 0.972 |

IDs (0, 15, 16) demonstrate that our surrogate task is robust to image sampling resolutions, as downsampling to $\frac{1}{1.6}$ or even $\frac{1}{4}$ has minimal impact on the results. Moreover, ID(13) further confirms that MIR remains effective even when trained on small-scale datasets (fewer than 1k samples). These insights provide potential opportunities for applying SD models to other dense estimation tasks and enhancing result sharpness. **In summary,** these ablation results validate the effectiveness of our proposed adaptation protocol, indicating that each design plays a crucial role in optimizing the diffusion model for self-supervised depth estimation tasks.

## 4.5 Inference Latency

As shown in the table below (MACs and Runtime are measured on a image with 1024×320 resolution on RTX 4090.), while Jasmine is more computationally expensive than prior self-supervised methods, it follows the trend of models like Marigold in trading cost for superior performance. Notably, our SSG module adds negligible latency, with Jasmine's runtime being comparable to Lotus.

| Method | Marigold | Lotus | E2E-Mono | Monodepth2 | MonoViT | MonoViFi | Jasmine |
|---|---|---|---|---|---|---|---|
| MACs | 133T | 2.65T | 2.65T | 21.43G | 25.63G | 28.79G | 2.83T |
| Runtime | 9.88s | 157ms | 152ms | 33ms | 29ms | 25ms | 172ms |

## 5 Conclusion

We propose Jasmine, the first SD-based self-supervised framework for monocular depth estimation, effectively leveraging SD's priors to enhance sharpness and generalization without high-precision supervision. To achieve this objective, we introduce two novel modules: Mix-batch Image Reconstruction (MIR) for mitigating reprojection artifacts and preserving Stable Diffusion's latent priors, alongside Scale-Shift GRU (SSG) to align scale-invariant depth predictions while suppressing noisy gradients. Extensive experiments demonstrate that Jasmine achieves SoTA performance on KITTI and superior zero-shot generalization across datasets. Our approach establishes a new paradigm for unsupervised depth estimation, paving the way for future advancements in self-supervised learning.

## Acknowledgment

This work was supported by the National Natural Science Foundation of China (NSFC) under Grant (U2441242,62172032) and Graduate Research Innovation Project under Grant (KKYJS25001536).

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

# Appendix

In this appendix, we provide more implementation details, experiments, analysis, and discussions for a comprehensive evaluation and understanding of Jasmine. Detailed contents are listed as follows:

# Contents

## A   Proof of Scale-Invariant Depth

For SSI depth, we assume that the estimated depth $\hat{D}$ and the ground truth depth $D$ have the following relationship:

$$\hat{D} = s_c D + s_h$$

Meanwhile, the depth from another viewpoint can be denoted as $g_1(D')$. Here, $g_1$ represents a transformation, and it is straightforward to see that this must be a linear transformation since the known identity only contains linear terms. Introducing a nonlinear transformation would not hold for the depth of arbitrary scenes. Thus, we can define:

$$g_1(D') = a_1 D' + b_1$$

Similarly, we can define the transition as $g_2(\mathbf{T})$, which is also a linear transformation, given by:

$$g_2(\mathbf{T}) = a_2 \mathbf{T} + b_2$$

Note that for transformations within the same scene, the relative pose $\mathbf{R}$ remains constant[18]. Also, $a$ and $b$ here can be arbitrary numbers, so we have:

$$g(x) - x \simeq g(x) - a_3 \simeq g(x) \tag{13}$$

Table 4: **Additional quantitative results on the KITTI dataset.**

| Method | Venue | Notes | Data | AbsRel | SqRel | RMSE | RMSElog | $a_1$ | $a_2$ | $a_3$ |
|---|---|---|---|---|---|---|---|---|---|---|
| ManyDepth[55] | CVPR2021 | ( -1,0) | K(40K+0) | 0.091 | 0.694 | 4.245 | 0.171 | 0.911 | 0.968 | 0.983 |
| Dual Refine[1] | CVPR2023 | (-1,0) | K(40K+0) | 0.087 | 0.674 | 4.130 | 0.167 | 0.915 | 0.969 | 0.984 |
| Mono-ViFI[29] | ECCV2024 | (-1,0,1) | K(40K+0) | 0.089 | **0.556** | 3.981 | **0.164** | 0.914 | 0.971 | **0.986** |
| EPCDepth[36] | ICCV2021 | Stereo | K(45K+0) | 0.091 | 0.646 | 4.207 | 0.176 | 0.901 | 0.966 | 0.983 |
| PlaneDepth[52] | CVPR2023 | Stereo | K(45K+0) | **0.085** | 0.563 | 4.023 | 0.171 | 0.910 | 0.968 | 0.984 |
| Jasmine | - | Mono | KH(68K+0) | 0.090 | 0.581 | **3.944** | **0.161** | **0.919** | **0.972** | **0.986** |

Assuming the SSI depth is work, we have:

$$\zeta' g_1(D') = K \left( \mathbf{R} K^{-1} \zeta(s_c D + s_h) + g_2(\mathbf{T}) \right) \tag{14}$$

Eq. 7 in the paper is:

$$\zeta' s_c D' = K \left( \mathbf{R} K^{-1} \zeta s_c D + s_c \mathbf{T} \right), \tag{15}$$

Subtracting Eq. 15 from Eq. 14, we have:

$$\zeta' \left( g_1(D') - s_c D' \right) = K \left( \mathbf{R} K^{-1} \zeta s_h + (g_2(\mathbf{T}) - s_c \mathbf{T}) \right)$$

From Eq. 13, it can be simplified as:

$$\zeta' g_1(D') = K \left( \mathbf{R} K^{-1} \zeta s_h + g_2(\mathbf{T}) \right)$$

Multiplying both sides by $K^{-1}$, we get:

$$K^{-1} \zeta' g_1(D') = \mathbf{R} K^{-1} \zeta s_h + g_2(\mathbf{T})$$

This is Eq. 8 in the paper.

## B  Detailed Training Workflow and Pseudocode

### B.1  Step-by-Step Workflow

In this section, we temporarily omit the specifics of MIR and SSG to clarify the integration of self-supervision with the Stable Diffusion framework in Jasmine:

- **Input:** A sequence of temporally adjacent images (like video frames), e.g., a source image $I_{t'}$ and a target image $I_t$.
  - Here $I_t$ is exactly the training image, $I_{t'}$ can be the previous frame $I_{t-1}$ or the next frame $I_{t+1}$ of $I_t$.

- **Step 1: Depth Prediction (via SD U-Net):**
  - The SD U-Net, $f_\theta^z$, takes the latent of the target image, $z^I = \text{VAE.encode}(I_t)$, and a pure noise vector, $\mathbf{n}$, as input.
  - It performs a single-step denoising process to predict the latent representation of a depth map, $z_0^y$.
    - ∗ This is the key point we mentioned in Sec 3.1: single-step denoising make this step become a fast, feed-forward process rather than computationally prohibitive with slow, iterative denoising.
  - The VAE decoder then converts $z_0^y$ into the final depth map, $D = \text{VAE.decode}(z_0^y)$.
  - **Crucially, no GT depth is ever used in this step.**

- **Step 2: Pose Prediction:**
  - A separate PoseNet takes the image pair $(I_t, I_{t'})$ as input and predicts the relative camera pose (rotation and translation), $T_{t \to t'}$.

- **Step 3: Self-Supervised Signal Generation (Image Reprojection[48]):**
  - Using the predicted depth map $D$ and the predicted pose $T_{t \to t'}$, we perform a warping operation to reproject the pixels from the source image $I_{t'}$ onto the target image's coordinate system. This creates a synthesized target image, $\hat{I}_t$.

- **Step 4: Loss Calculation and Optimization:**

- A photometric reprojection loss, $L_{ph}$, is calculated by comparing the synthesized image $\hat{I}_t$ with the original target image $I_t$.
- **This loss, $L_{ph}$, is the core self-supervised signal.** If the predicted depth $D$ is incorrect, the reprojected image $\hat{I}_t$ will not match the original $I_t$, resulting in a high loss.

- **Step 5: End-to-End Backpropagation:**
  - The gradient from $L_{ph}$ is backpropagated through the entire computational graph. This means the gradient flows back to update the weights of both the **PoseNet** and, most importantly, the **SD U-Net** ($f_\theta^z$).

- **Output:** The predicted depth map $D$ obtained from Step 1.

### B.2 Training Pseudocode

Algorithm 1 outlines the complete training procedure for Jasmine, incorporating all components including MIR, SSG, and the full loss computation (Corresponds to the pipeline in Fig. 2).

---

**Algorithm 1** Jasmine Training Algorithm

---

1: **Initialize:** SD U-Net, PoseNet, SSG, VAE, optimizer
2: VAE.eval()                                                                   ▷ VAE weights are frozen
3: **for** each batch of inputs 'I' in dataloader **do**
4:                                                              ▷ **Mix-batch Image Reconstruction (MIR)**
5:      $s_{depth} \leftarrow [1, 0]$, $s_{recon} \leftarrow [0, 1]$                                      ▷ Task switchers
6:      $z_t^I, z_m^I, z^n \leftarrow$ VAE.encode(I['I_t'], I['I_m'], noise)
7:      $D_{ssi} \leftarrow$ VAE.decode(SD_UNet($[z_t^I, z^n], t = 999, s = s_{depth}$))
8:      $I_{rec} \leftarrow$ VAE.decode(SD_UNet($[z_m^I, z^n], t = 999, s = s_{recon}$))
9:                                                                     ▷ **Scale-Shift GRU (SSG)**
10:      $D_{list} \leftarrow$ SSG($D_{ssi}, z_t^I$)                             ▷ $D_{list}$ contains $[D_{ssi}, D_1, D_{si}]$
11:      $pose \leftarrow$ PoseNet(I['I_t'], I['I_t'])                       ▷ Pose Estimation for Self-Supervision
12:      $loss \leftarrow$ compute_loss(I, $D_{list}$, pose, $I_{rec}$)            ▷ Loss Computation and Optimization
13:      optimizer.zero_grad()
14:      $loss$.backward()
15:      optimizer.step()
16: **end for**
17: **function** COMPUTE_LOSS(inputs, $D_{list}$, pose, $I_{rec}$)
18:      $D_{si}, D_{ssi} \leftarrow D_{list}[-1], D_{list}[0]$
19:      $reproj\_img \leftarrow$ reproject(inputs['I_t'], $D_{si}$, pose)
20:      $L_{ph} \leftarrow$ photometric_loss($reproj\_img$, inputs['I_t'])          ▷ Core self-supervised signal
21:      $L_s \leftarrow$ photometric_loss($I_{rec}$, inputs['I_m'])                  ▷ Surrogate task signal
22:      $L_{tc} \leftarrow$ teacher_loss($D_{list}$, inputs['D_tc'])                    ▷ As per Eq. 11
23:      $L_a \leftarrow$ compute_auxiliary_loss(inputs, $D_{si}, D_{ssi}$)
24:      $loss \leftarrow L_{ph} + L_s + L_{tc} + L_a \cdot 0.008$
25:      **return** $loss$
26: **end function**
27: **function** COMPUTE_AUXILIARY_LOSS(inputs, $D_{si}, D_{ssi}$)
28:      $L_{GDS} \leftarrow$ gds_loss(inputs['I_t'], inputs['seg'], $D_{si}$)
29:      $L_{SKY} \leftarrow$ sky_loss($D_{ssi}$, inputs['sky_mask'])
30:      $L_e \leftarrow$ edge_loss($D_{si}, D_{ssi}$)
31:      **return** $L_{GDS} + L_{SKY} + L_e$
32: **end function**

---

## C  Supervision Details

Self-supervised depth estimation has been researched for a decade, and hundreds of works have proposed numerous progressive ideas. Therefore, to achieve SoTA performance, it is inevitable that we will reuse some loss constraints from previous works to optimize our results. In the following text, except for the **edge loss**, the other losses are mostly referenced from prior papers and are not the core contributions of this paper. Thus, we have not included them in the main paper. The effectiveness of

the other papers' losses has been comprehensively proved in their papers, and we did not conduct additional ablation studies for them. For the loss specific in this paper, we present some visualized ablation results in Fig. 7.

## C.1    SGD Loss for Dynamic Object

We first introduce the most commonly used smoothness loss $L_{sm}$ in self-supervised depth estimation, which encourages locally smooth depth maps while preserving edges in the image. Its specific expression is as follows:

$$L_{sm} = |\partial_x \hat{D}|e^{-|\partial_x \mathbf{I}|} + |\partial_y \hat{D}|e^{-|\partial_y \mathbf{I}|}, \tag{16}$$

where $\hat{D}$ is the depth normalized by the mean of $D$.

Building on this, we adopt the GDS loss (only the first stage) from[34] to handle dynamic objects. This approach is based on the smoothness loss and introduces a ground-contact-prior mask $M_{gr}$, defined as:

$$M_{gr}(i,j) = \gamma \cdot M_t(i,j) + (1 - M_t(i,j))$$

where $M_t$ is the dynamic object segmentation obtained from a semantic segmentation network (1 represents dynamic and 0 for static), and $\gamma$ is the weighting parameter for $M_{gr}$, empirically set to 100. Considering the bottom pixels of dynamic regions like the car tire, they impose a high weighting on $|\partial_y \hat{d}_t|$ with $M_{gr}(i,j) = \gamma$, thereby enforcing its depth consists with its neighboring ground pixels below. So the final loss is:

$$L_{GDS} = |\partial_x \hat{D}|e^{-|\partial_x \mathbf{I}|} + |\partial_y \hat{D}|M_{gr}e^{-|\partial_y \mathbf{I}|}. \tag{17}$$

## C.2    Edge Loss for Sharp Prediction

We further introduce an edge loss to enhance the prediction's details. Since the surrogate and primary tasks are decoupled after the U-Net's final layer, and the Scale-Shift Invariant (SSI) depth is shielded from photometric loss interference through SSG-based isolation, the SSI depth retains significantly richer structural details. To transfer these details to the final result, we introduce a simple edge loss. Specifically, we design a GradNet,

$$\text{GradNet}(x) = (|\text{conv2d}(x, w_x)|, |\text{conv2d}(x, w_y)|),$$

where $w_x = [[-1,0,1],[-2,0,2],[-1,0,1]]$ and $w_y = [[-1,-2,-1],[0,0,0],[1,2,1]]$ are the convolutional kernels for computing gradients in the $x$ and $y$ directions. Subsequently, the edge loss is defined as:

$$L_e = L_1(C_x, D_x) + L_1(C_y, D_y), \tag{18}$$

where $(C_x, C_y) = \text{GradNet}(\hat{D}_{\text{SSI}})$ represents the normalized gradient of the detached SSI depth, and $(D_x, D_y) = \text{GradNet}(\hat{D}_{\text{blur}})$ represents the gradient of blur depth. We implement this edge loss to the SSG outputs $D_1$ and SI depth $D_2$. The detached $\underline{D_{\text{SSI}}}$, while the depth distribution is inaccurate, has sharper edges, making it an excellent teacher. The prediction comparison with and without edge loss are shown in Fig. 7 (d-4) and (d-3), respectively.

## C.3    Sky Loss for Anti-artifact

Our experiments reveal that (Fig. 7 (b-2)), although the edge loss significantly enhances the model's details, artifacts still appear outside object edges, particularly in the sky. This is because the sky is a texture-less region, causing self-supervised models to produce erroneous estimates. However, this typically does not affect performance, as during testing, the sky is either cropped (Eigen Crop) or its ground truth depth is invalid and excluded from evaluation. To address this issue, we introduce a sky loss:

$$L_{\text{sky}} = \eta_{\text{sky}} \cdot L_1(D, D_{\text{max}} \cdot \mathbf{1}_D, M_{\text{sky}}), \tag{19}$$

where $D$ is the predicted depth, $D_{\text{max}}$ is the maximum depth value, $\mathbf{1}_D$ is a tensor of ones with the same shape as $D$, and $M_{\text{sky}}$ is the sky mask derived from the semantic segmentation. $\eta_{\text{sky}}$ is a weighting factor set to 0.1.

Note that the sky loss does not add extra details. As shown in Fig. 7 (b-2) and (b-3), the original details are already captured by the model; the sky loss merely sets the sky depth to infinity.

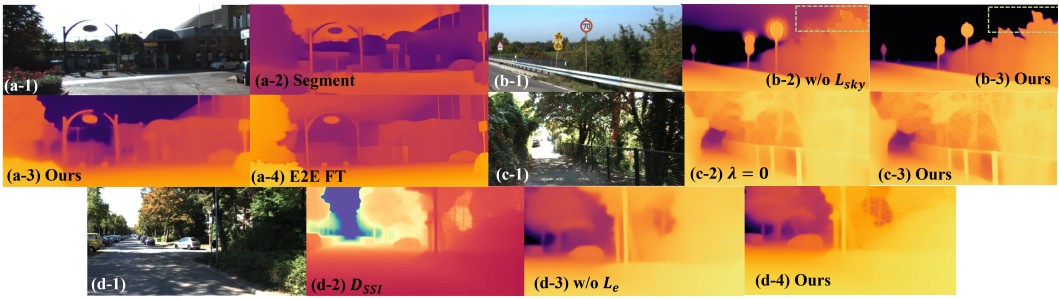

Figure 7: **Qualitative Ablation Study of Adaptive Loss.** Notably, while (a-2) demonstrates superior visual quality, it exhibits an entirely inaccurate depth distribution. (a-4) is the result of E2E FT, which can serve as a pseudo-label here.

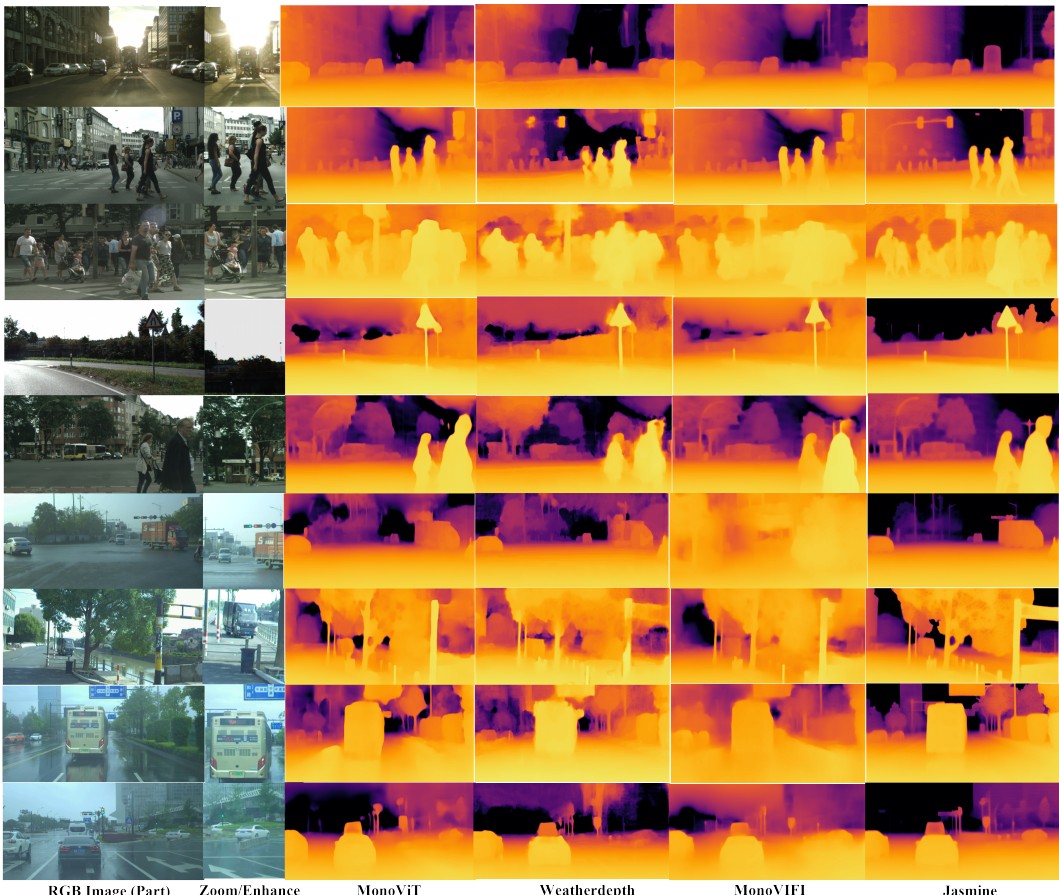

Figure 8: **Qualitative results** on zero-shot Scale-Invariant depth estimation.

## C.4 Eliminating the Bottleneck of Teacher Loss

In this section, we elaborate on the details of the implementation of teacher loss. We use MonoViT as the teacher model. The model estimates disparity $dp$, and thus we obtain the depth $D_{tc}$ through:

$$\frac{1}{D_{tc}} = \text{clip}\left(\frac{1}{D_{max}} + \left(\frac{1}{D_{min}} - \frac{1}{D_{max}}\right) \cdot dp, \, 0, \, 3\right),$$

where $D_{min}$ and $D_{max}$ are set to 0.1 and 100, respectivley. We find that the depth range at range [0,3] can already describe all information within the maximum depth.

For the SSI depth supervision, we perform a normalization similar to Eq. 9 in paper:

$$\text{Norm}(D_{\text{tc}}) = 2 \cdot \left( \frac{D_{\text{tc}} - \min(D_{\text{tc}})}{\max(D_{\text{tc}}) - \min(D_{\text{tc}})} - 0.5 \right). \tag{20}$$

Furthermore, to avoid the performance ceiling imposed by the teacher model, we draw inspiration from D4RD[50], and employ an adaptive filtering mechanism:

$$M_{tc} = \left[ \min_{t'} ph\left(\mathbf{I}, \mathbf{I}_{t' \rightarrow t}^{tc}\right) < \frac{\lambda}{\eta_{\text{step}}} \right],$$

where $\lambda$ is a constant set to 1.5, $\mathbf{I}_{t' \rightarrow t}^{tc}$ is the warped target image using the teacher disparity $D_{tc}$, and $\eta_{\text{step}} = \max(1, 30 \cdot (\text{step}_{now}/\text{step}_{max}))$ is a dynamic factor that adjusts with training progress. This adaptive weight initially allows the model to converge across the entire depth map and subsequently filters out less accurate regions, mitigating the adverse effects of inaccurate pseudo-depth labels. Therefore, we have:

$$\text{filter}\left(D_{\text{tc}}\right) = D_{\text{tc}} \cdot M_{tc},$$

In the supervision process, we introduce the BerHu loss $L_B$ and get better results:

$$L_B(x, y) = \begin{cases} |x - y|, & \text{if } |x - y| \leq c, \\ \frac{(x-y)^2 + c^2}{2c}, & \text{if } |x - y| > c, \end{cases} \tag{21}$$

where $c = 0.2 \cdot \max(|x - y|)$ is a threshold that adapts to the error magnitude. This loss imposes a greater penalty on pixels with larger errors using an $L_2$-like penalty, while retaining the robustness of $L_1$ loss for small errors.

Thus, The Eq. 13 in paper:

$$\begin{aligned} L_{tc} = (&L_B\left(\text{norm}\left(D_{tc}\right), D_{\text{SSI}}\right) + L_B\left(D_{tc}, D_1\right) \cdot \eta_{t1} \\ &+ L_B\left(\text{filter}\left(D_{tc}\right), D_{SI}\right) \cdot \eta_{t2}\right)/\eta_{\text{step}}, \end{aligned} \tag{22}$$

is fully detailed in this part. The teacher loss not only stabilizes our model but also avoids limiting its performance potential.

**The segmentation model** mentioned above is Mask2Former[6] throughout. To avoid the misconception that the details stem from the segmentation network's output, we further expand Fig. 3 (in paper) with segment-based methods. In fact, to preserve these details, a naive approach is to employ semantic segmentation constraints[34]. However, as shown in Fig. 7 (a-2), using the semantic triplet loss from [34] not only disrupts the depth distribution but also introduces spurious edges, rendering it incompatible with SD's latent priors. Furthermore, the results in paper Table 1 are compared with a segmentation-based model Jaeon et al*, and our method achieves superior performance.

# D  Preliminaries of GRU

The Gated Recurrent Unit (GRU) is a type of recurrent neural network (RNN) that uses gating mechanisms to control the flow of information between the previous hidden state and the current input, making it effective for sequence modeling tasks. Recently, GRUs have been gradually used in depth estimation[45, 43]. A GRU cell updates its hidden state $\mathbf{h}^{k+1}$ based on the previous hidden state $\mathbf{h}^k$ and the current input $\mathbf{x}^k$. This update process is primarily managed by two gates: the **reset gate** ($\mathbf{r}$) and the **update gate** ($\mathbf{z}$). The reset gate determines how much of the past information (from the previous hidden state) to effectively "forget" or "reset" before computing a new candidate hidden state. The update gate then decides how much of this new candidate hidden state should be incorporated into the final hidden state, versus how much of the previous hidden state to carry over. The mathematical formulation for these operations is as follows:

$$\begin{aligned} \mathbf{z}^{k+1} &= \sigma([\mathbf{h}^k, \mathbf{x}^k], W_z), \quad \mathbf{r}^{k+1} = \sigma([\mathbf{h}^k, \mathbf{x}^k], W_r), \\ \widehat{\mathbf{h}}^{k+1} &= \tanh(Conv([\mathbf{r}^{k+1} \odot \mathbf{h}^k, \mathbf{x}^k], W_h)), \\ \mathbf{h}^{k+1} &= (1 - \mathbf{z}^{k+1}) \odot \mathbf{h}^k + \mathbf{z}^{k+1} \odot \widehat{\mathbf{h}}^{k+1}, \end{aligned} \tag{23}$$

where $k$ denotes iteration steps, $\sigma$ is the sigmoid activation function (outputting values between 0 and 1, ideal for gating), and $\odot$ indicates element-wise multiplication. The terms $W_z, W_r, W_h$ represent the learnable parameters (typically weight matrices and biases) for the update gate, reset gate, and

candidate hidden state computation, respectively. The $Conv$ notation suggests that convolutional layers are used for these transformations, as is common when applying GRUs to feature maps in computer vision tasks. In our paper, the refined hidden state $\mathbf{h}^{k+1}$ predicts depth adjustment $D_\delta$ to update $D_k$, yielding $D_{k+1}$ for subsequent iterations, as utilized in our Scale-Shift GRU (SSG) module (see Sec. 3.3).

# E   Addition Experiments Results

## E.1   Evaluation on KITTI Improved Benchmark

The standard self-supervised evaluation on the KITTI dataset is typically conducted using the raw LiDAR GT. However, due to the noisy nature of LiDAR data and known preprocessing issues[1], we also provide the eigen improved benchmark result. Following the practice adopted in Mono-ViFi [29], as shown in Table 5, this evaluation further confirms Jasmine's SoTA performance, where it consistently outperforms other leading methods.

Table 5: **Quantitative results on the KITTI dataset using the improved GT [47].** The performance of SD-based methods is not fully reported in their original papers.

| Method | AbsRel↓ | SqRel↓ | RMSE↓ | RMSElog↓ | $a_1$↑ | $a_2$↑ | $a_3$↑ |
|---|---|---|---|---|---|---|---|
| Marigold [25] | 0.099 | - | - | - | 0.916 | 0.987 | - |
| E2E-FT [33] | 0.096 | - | - | - | 0.921 | 0.980 | - |
| Lotus [21] | 0.081 | - | - | - | 0.931 | 0.987 | - |
| Jasmine* | **0.064** | **0.294** | **2.982** | **0.097** | **0.957** | **0.994** | **0.998** |
| MonoViT [83] | 0.068 | 0.314 | 3.125 | 0.105 | 0.948 | 0.992 | 0.998 |
| MonoViFi [29] | 0.071 | 0.338 | 3.539 | 0.113 | 0.937 | 0.990 | 0.998 |
| Jasmine | **0.061** | **0.255** | **2.765** | **0.092** | **0.963** | **0.995** | **0.999** |

## E.2   Comparison with Other Self-Supervised Settings

As mentioned in Sec. 2, our single-frame monocular approach is more challenged but practical compared to other self-supervised configurations. Compared to stereo-based methods, monocular methods face additional challenges with dynamic objects and pose estimation inaccuracy, but monocular methods can eliminate the need for synchronized binocular cameras and precise calibration. Similarly, single-frame models neglect temporal information during inference, while multi-frame methods leverage consecutive frames to construct cost volumes and even support iterative refinement at test time, yielding improved accuracy. However, the practical deployment of multi-frame methods remains constrained by the availability of multiple frames.

Despite these inherent disadvantages, as shown in Table 4, our method still outperforms the state-of-the-art approaches in both multi-frame and stereo-supervision domains across most metrics. This remarkable achievement, especially considering our more challenging problem setting, further demonstrates the substantial strength and generalizability of our approach.

## E.3   Qualitative Comparisons

In Fig. 8, we further compare the performance of our Jasmine with other methods in multi scenes. The quantitative results obviously demonstrate that our method can produce much finer and more accurate depth predictions, particularly in complex regions with intricate structures, which sometimes cannot be reflected by the metrics.

## E.4   Mix-batch Ratio Ablation Details

In the paper, Fig. 3 (g) illustrates the performance variation versus the mix-batch ratio under two supervision schemes. We further present

Table 6: **Ablation Studies** on the mix-batch ratio. Ph refers to supervision using Eq. 5, while Latent refers to supervision using Eq. 4 . The notation $+\lambda$ denotes the proportion of the KITTI dataset used (*e.g.*, $+0.3$ indicates a KITTI:Hypersim ratio of 3:7 in the MIR training data.)

| (ID) Method | AbsRel | SqRel | RMSE | $a_1$ | $a_2$ |
|---|---|---|---|---|---|
| Ph+0 | **0.089** | **0.573** | 3.973 | 0.918 | 0.972 |
| Ph+0.3 | 0.090 | 0.581 | 3.944 | **0.919** | 0.972 |
| Ph+0.6 | 0.092 | 0.593 | **3.933** | 0.915 | **0.973** |
| Ph+1 | 0.093 | 0.590 | 3.970 | 0.915 | **0.973** |
| Latent+0 | 0.106 | 0.614 | **4.181** | 0.901 | 0.970 |
| Latent+0.3 | **0.095** | 0.606 | 4.138 | **0.909** | 0.970 |
| Latent+0.6 | 0.121 | 0.649 | 4.322 | 0.876 | **0.970** |
| Latent+1 | 0.129 | 0.679 | 4.385 | 0.858 | 0.962 |

---

[1]Discussed in MonoDepth2 repository: https://github.com/nianticlabs/monodepth2/issues/274

the complete results in Table 6. Note that although "Ph+0" offers better metrics, it predicts blurred results (7 (c-2)).

## F  Analysis of SD Prior Degradation via Self-Supervision

For clarity, we first revisit the core definition of disparity: when capturing two images of the same scene from different camera positions, the **same point** will appear at different pixel coordinates in each image. This difference is known as *disparity*. In fact, disparity and depth are inversely proportional and correspond one-to-one.

From this perspective, the loss function in Eq. 1 is designed to find, for each point in the target view $I_t$, the most **similar point** in the source view $I_{t'}$. From the coordinate difference between these points $I_{t' \to t}$, we obtain the depth supervision (Eq. 2). However, if the corresponding point in $I_t$ is occluded in $I_{t'}$, the optimization process is forced to select the "least bad" alternative, resulting in an incorrect match and, consequently, erroneous depth estimation.

Taking Fig. 3 (a) as an example, the scene consists of a infinite background (depth=$\infty$), an orange rectangle (20m), and a light blue tree (10m). Here, the camera only shifts horizontally between the target and source views. For point q (tree), the correct disparity is 10 pixels (10m). For point p,r (rectangle), the correct disparity should be 5 pixels (20m).

Due to camera movement, the tree in the source view occludes the correct matching point of p. To minimize photometric loss, the algorithm searches for the most similar region nearby and once again finds the orange area at p', which is 10 pixels (10m).

This error causes the expected depth edge on the right side of point p to disappear (right side's depth are all 10m, reason same to p), resulting in a blurred boundary in the depth map. When such depth map is used as a supervisory signal to guide the SD model, it effectively introduces "noisy" data. This forces the SD model to learn and reproduce these incorrect and blurred boundaries, thereby undermining its strong prior knowledge of clear object boundaries.

## G  Evaluation Metrics

Similar to [15], we employ the following evaluation metrics in our experiments,

AbsRel: $\frac{1}{|M_{vl}|} \sum_{d \in M_{vl}} |d - d_{gt}| / d_{gt}$;

SqRel: $\frac{1}{|M_{vl}|} \sum_{d \in M_{vl}} \|d - d_{gt}\|^2 / d_{gt}$;

RMSE: $\sqrt{\frac{1}{|M_{vl}|} \sum_{d \in M_{vl}} \|d - d_{gt}\|^2}$;

RMSElog: $\sqrt{\frac{1}{|M_{vl}|} \sum_{d \in M_{vl}} \|\log(d) - \log(d_{gt})\|^2}$;

$a_t$: percentage of $d$ such that $\max(\frac{d}{d_{gt}}, \frac{d_{gt}}{d}) < 1.25^t$ ; where $d_{gt}$ and $d$ denote the GT and estimated pixel depth, $M_{vl}$ is the valid mask set to $1e - 3 < d_{gt} < 80$.

## H  Error Bar Analysis

As highlighted in Sec 3.4, directly training Jasmine can be unstable due to the SD's enormous size, joint training across modules, and indirect self-supervisory mechanisms. To further ensure stability, we also implement a module-wise freezing training strategy, which involves 3 key phases: In the beginning, we enable gradient updates for all network components. Subsequently, we freeze the SSG and Posenet modules to decouple depth-pose optimization while maintaining fixed parameters. After achieving convergence to a suboptimal solution, we reintroduce gradient updates to these modules for final optimization. In the training procedure, the first training phase involved 15k steps,

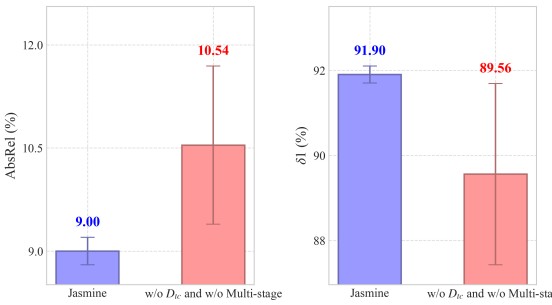

Figure 9: **Error Bar Analysis on KITTI Eigen test split**. We conduct this analysis through multiple training runs and observe the performance oscillations after 25k steps.

while the second and third phases automatically transitioned, sharing an additional 10k steps for a total of 25k training steps. As illustrated in Fig. 9, we demonstrate that the pseudo-label supervision and module-wise freezing training are particularly crucial for steady training in complex, multi-module self-supervised systems.

# I Limitation and Future Work

As noted in Sec. 2, the ubiquitously available videos suggest that the self-supervised methods possess significant data advantages and working potential. In this paper, Jasmine is trained on only tens of thousands of data samples and only on a driving dataset (KITTI), leaving room for further exploration in scaling up training data to other domains (industry, indoor, etc). We believe that if there really emerges a 'GPT' moment for 3D perception in the future, it will more likely involve self-supervised methods trained on videos rather than learning from annotated depth. Furthermore, we believe that the unsupervised Stable Diffusion fine-tuning paradigm proposed in this paper can be applied to other related fields, such as depth completion [65, 63, 66, 70, 71, 60], depth super-resolution [64, 54], Multi-view Stereo [74, 75, 72, 73], Stereo Matching [56, 58, 57, 53] and Language Aid Depth Estimation [78, 80, 79, 82].

