# OpenReview forum: "Jasmine: Harnessing Diffusion Prior for Self-supervised Depth Estimation"
_NeurIPS.cc/2025/Conference — NeurIPS 2025 poster_

### Official Review · Reviewer_s3Km · 2025-06-27

**Clarity:** 3
**Significance:** 2
**Originality:** 2
**Rating:** 4
**Confidence:** 5

**Summary:**

The manuscript proposes Jasmine, the Stable Diffusion (SD) based self-supervised monocular depth estimation framwork that utilizes SD’s visual priors to enhance the sharpness and generalization of unsupervised prediction. Jasmine leverages a mix-batch image reconstruction task to preserve the detailed latent priors of SD models. Additionally, a Scale-Shift GRU is introduced to for learning the scale misalignment from the pseudo-lables.

**Questions:**

•	Provide a detailed literature review to substantiate the novelty of the mix-batch reconstruction and Scale-Shift GRU (how SST works).

•	Include ablation studies to validate $L_{GDS}$ and $L_{SKY}$, comparing their impact to existing techniques for dynamic objects and noise mitigation.

•	Present a comprehensive comparison across multiple datasets (e.g., Cityscapes, Waymo Open) and resolutions.

•	Evaluate performance with other teacher models (e.g., monodepth2, LiteMono and Mono-VIFI) and without a teacher model to demonstrate the method’s independent effectiveness.

•	Report computational requirements and inference latency, comparing Jasmine to other models.

**Ethical Concerns:**

["NO or VERY MINOR ethics concerns only"]

**Final Justification:**

I think the authors have addressed my concerns to some extent, I would like to upgrade my rating as borderline accept

**Limitations:**

please refer to the weakness

**Quality:**

2

**Strengths And Weaknesses:**

Paper Strengths:

1.	The introduction is clear in describing the addressed problem and describing the proposed method. Differences to existing work are clearly discussed in the related work section.

2.	The method description is clear with detailed mathematical descriptions for the proposed loss components.

3.	The proposed Jasmine reprensents an interesting and structured approach to levrage Stable Diffusion model’s visual priors to obtain fine-grained depth prediction.


Paper Weaknesses:

1.	The proposed mix-batch image reconstruction task appears to be a straightforward adaptation of Stable Diffusion’s Img2Img pipeline, with minimal modifications for depth estimation. Similarly, the use of synthetic images as training inputs directly mirrors strategies in Marigold [1] Ke B, Obukhov A, Huang S, et al. Repurposing diffusion-based image generators for monocular depth estimation[C]//Proceedings of the IEEE/CVF Conference on Computer Vision and Pattern Recognition. 2024: 9492-9502, offering little to no novel contribution to the field.

2.	The paper employs $L_{GDS}$ from [2] Moon J, Bello J L G, Kwon B, et al. From-Ground-To-Objects: Coarse-to-fine self-supervised monocular depth estimation of dynamic objects with ground contact prior[C]//Proceedings of the IEEE/CVF Conference on Computer Vision and Pattern Recognition. 2024: 10519-10529. and $L_{SKY}$ as part of the proposed framework but fails to include ablation studies to quantify their impact on performance. While $L_{GDS}$ reportedly enhances performance on KITTI by handling dynamic objects, and $L_{SKY}$ may reduce noise. Their discussion is limited to the appendix, reducing novelty and transparency and the lack of empirical validation obscures the method’s contributions.

3.	The training process requires sequential optimization of multiple networks (Stable Diffusion model, Scale Shift Transformer and Sep-Conv GRU and Posenet) with numerous loss functions ($L_s$, $L_{ph}$, $L_{tc}$ and $L_{GDS}$, $L_{SKY}$, $L_{e}$) and multiple switch overs. This complexity may hinder reproducibility (no code available) and deployment.

4.	While Stable Diffusion models offer rich visual priors, their large model size introduces significant computational burdens and high inference latency, which the paper fails to address explicitly. This omission limits the fairness of comparisons with lighter models (e.g., CNN-based self-supervised methods) and obscures practical deployment constraints.

---

> ### Author Rebuttal · Authors · 2025-07-30
>
> # Overview
>
> We thank you for your feedback. We appreciate that you recognized our work as **"clearly described"** and an **"interesting and structured approach."** Regarding the concerns raised, we found the feedback on our ablation studies and computational cost analysis invaluable for strengthening the paper. However, on the core assessment of our novelty/transparency, we must respectfully beg to differ. Due to space constraints, some tables below are included in other reviewers' rebuttals, and the ablation table only highlights essential metrics.
>
> # On Novelty: Differentiating from Prior Art
>
> > Weaknesses 1.
> > Questions 1.
>
> Jasmine is the **first** SD-based SSMDE framework, and our contribution can be located at Lines 65-71. **For the MIR:**
>
> 1. We have made numerous modifications: within each training batch, the same SD model alternately reconstructs synthesized or real images, as well as predicts depth maps through a switcher; the reconstructed images are supervised using a photometric loss rather than the traditional latent MSE loss.
> 2. Our motivations also differ fundamentally from those of SD: Self-supervision tends to erode the prior knowledge embedded in SD, making it unsuitable for direct application as in Marigold. Under this motivation, we introduced MIR to preserve this prior **, with important modifications for SSMDE**.
> 3. We conducted extensive ablation experiments on the use of auxiliary images, which yielded valuable insights. It is worth noting that MIR is only one aspect of our overall contribution.
>
> **For synthetic images usage,** Marigold must use synthetic data to keep its latent space intact, as noise-free, dense depth GT is extremely rare. In contrast, our Jasmine model employs real KITTI images for its depth training, and its auxiliary images can be flexibly selected from either real or synthetic sources.  This is because Jasmine's self-supervised process does not require high-quality GT, but supervised Marigold does. Therefore, we actually do **NOT** mirror Marigold, but a pioneering work that effectively integrates diffusion priors into SSMDE.
>
> In addition to MIR, the improvement of SSG is also the **first** work in self-supervision to analyze the difference between the two types of depth de-normalization. We provide a detailed analysis of these differences in Sec B, which, to our knowledge, is unprecedented in the literature. The SST module uses cross-attention to learn the scale/shift parameter from the last iteration prediction, as referred to the reviewer `SPog` rebuttal for more details.
>
> Finally, we note that the reviewer `arKR` highlighted our work as 'the first to integrate diffusion priors into SSMDE' and as 'establishing a new paradigm for the task.' Reviewer `SPog` remarked that our approach 'addresses a longstanding challenge in self-enhanced depth estimation,' and reviewer `fCDa` stated that we 'introduce a novel approach to harness the SD visual priors for SSMDE.' Collectively, these comments affirm that our work is innovative and makes meaningful contributions to the field.
>
> # Missing Ablation Studies
>
> > Weaknesses 2. Questions 2.
>
> We would like to clarify that $L_{GDS}$, $L_{e}$, and $L_{sky}$ are **NOT** the core contributions of our work, which is why they are presented in the appendix. SSMDE is a well-established field, and many of its challenges have already been addressed by prior research. The loss functions mentioned in the appendix are **mature, widely adopted solutions,** and our intention is **NOT** to reinvent these established methods.
>
> Second, we have already visualized the ablation results for $L_{sky}$ and $L_{e}$ in Appendix Fig. 7 (b-2, b-3, d-3, d-4). To further address the reviewer's concerns, we have also provided ablation results in Table G. Without these losses, Jasmine remains at the SoTA level.
>
> |Model|AbsRel⬇️|RMSE⬇️|δ1⬆️|
> |-|-|-|-|
> |w/o$L_{GDS}$|0.092|4.121|0.912|
> |w/o$L_{e}$|0.090|4.055|0.915|
> |w/o$L_{sky}$|0.091|3.990|0.917|
>
> `Table G.` Ablation study on $L_{GDS}$,$L_{sky}$ and $L_{e}$, evaluation rules are same with Table 1 in the paper.
>
> Finally, another experiment is designed to **compare the performance of our improved SD priors against traditional frameworks, both building upon these losses.** In fact, in our paper, Table 1 (Jaeho et al.\*) shows the performance of the MonoViT framework with these mature Losses. Our method consistently outperforms Jaeho et al.\*. Here, we further include a zero-shot performance comparison in Table H, which confirms that incorporating the improved SD prior significantly enhances the model's generalization. These results show that the performance depends mainly on our new paradigm, rather than on these mature Losses.
>
> |Model|DrivingStereo||CityScape||
> |-|-|-|-|-|
> ||AbsRel⬇️|δ1⬆️|AbsRel⬇️|δ1⬆️|
> |Jaeho et al.\*|0.143|0.833|0.132|0.821|
> |Jasmine|0.136|0.850|0.123|0.852|
>
> `Table H.`
>
> # Training complexity
>
> > Weaknesses 3.
>
> Due to the rebuttal policy, we regret that we are forbidden to release our full code at this time. However, to enhance transparency, we have provided detailed pseudocode in our rebuttal and commit to releasing our code upon paper acceptance.
>
> ```
> SD_UNet, PoseNet, SSG, VAE = init_models()
> optimizer = init_optim(...)
> VAE.eval()
> for inputs in dataloader:
>  #MIR
>  s=[[0,1],[1,0]]#switcher
>  z^I_t, z^I_m, z^n = VAE.encode(inputs['I_t'],inputs['I_m'],noise)
>  D_ssi = VAE.decode(SD_UNet([z^I_t,z^n], t=999,s=s[0]))
>  I_rec= VAE.decode(SD_UNet([z^I_m,z^n], t=999,s=s[1]))
>  #SSG
>  D_list = SSG(D_ssi, z^I_t) #D_list=[D_ssi,D_1,D_si]
>  #SSMDE Pose
>  pose = PoseNet(inputs['I_t'], inputs['I_(t')'])
>  #Loss
>  loss = compute_loss(inputs, D_list, pose,I_rec)
>  #Optim
>  loss.backward(), optimizer.step()
> def compute_loss(inputs,D_list, pose,I_rec):
>  D_si,D_ssi=D_list['D_si'],D_list['D_ssi']
>  reproj_img = reproject(inputs['I_(t')'],D_si, pose)
>  L_ph = ph_loss(reproj_img, inputs['I_t'])
>  L_s = ph_loss(I_rec, inputs['I_m'])
>  L_tc= tc_loss(D_list,inputs['D_tc'],type='berhu')#Eq 11
>  L_a= compute_L_a(inputs,D_si,D_ssi)
>  loss =L_ph + L_s + L_tc+L_a*0.008
>  return loss
> def compute_L_a(inputs,D_si,D_ssi):
>  L_GDS = gds_loss(inputs['I_t'],inputs['seg'],D_si)
>  L_SKY = sky_loss(D_ssi, inputs['sky_mask'])
>  L_e = edge_loss(D_si,D_ssi)
>  return L_GDS+L_SKY+L_e
> ```
> Our training process is fully **end-to-end**. While multiple network modules are involved, only **3** core components require optimization: the U-Net of Stable Diffusion, PoseNet, and our proposed SSG. All modules are jointly updated using a single optimizer `optimizer.step()`, as illustrated in the provided pseudocode.
> Although there are multiple loss functions, they are all based on well-established and open-source prior works in the field(`ph_loss-> Monodepth2, sgd_loss->From Ground to Objects). This design choice ensures that our method remains conceptually straightforward to implement.
> With the provided pseudocode and clear references to existing methods, we believe our approach is logically transparent.
>
> # Computational costs and actual deployment
> > Weaknesses 4. Question 5.
>
> We acknowledge that SD-based methods introduce higher computational overhead, inference latency and affect the fairness of comparisons with traditional CNN-based SSMDE. But we argue this has a limited impact on our contribution.
>
> First, we report the computational requirements and inference latency in Table B (refer to rebuttal for reviewer `arKR`), comparing Jasmine with both self-supervised and SD-based models.
>
> Second, we wish to position our work in current MDE research trends. Recently, a series of leading works represented by **Marigold (CVPR 2024 Best Paper Candidate), Lotus(ICLR 2025) and DepthAnything (CVPR/NIPS 2024)** have established a research paradigm within the community: **there is a willingness to accept a certain level of computational cost at this stage in exchange for the unprecedented generalization and quality afforded by Large Vision Models(LVMs).** Our work shares the same philosophy as these studies. Therefore, the main contribution of our paper is not to pursue the ultimate inference speed, but to **explore and successfully introduce the powerful prior of SD into an SSMDE framework for the first time, aiming to solve the long-standing problems** of weak generalization and poor detail recovery in this field.
>
> Third, for **Application Potential**, the rapid adoption of Marigold by 30+ downstream tasks demonstrates that **even computationally intensive models can find widespread application if their performance is significant enough.** Similarly, we believe Jasmine, which does not require depth GT, has strong potential for real-world applications where annotated data is limited but leveraging LVMs is desirable.
>
> # Others
> > Question 3.
>
> Tables A and E (see rebuttal for reviewers `arKR` and `fCDa`) provide a detailed performance comparison at 640×192 and 1216×352 resolutions. Results on the Cityscapes dataset are already reported in Table 2 of the main paper. Since we have performed zero-shot evaluations on several street-view datasets, we believe that additional testing on a similar Waymo dataset (less used in this community) would offer limited insights. Instead, to better address your generalization concerns, we conducted evaluations on indoor datasets, as presented in Table C (see rebuttal for reviewers `arKR`).
>
> > Question 4.
>
> Table J reports the results when using monodepth2 as the teacher model, as well as the performance of our method without any teacher model. Due to time constraints, we are unable to perform comprehensive training with additional teacher models and choose the monodepth2 (a relatively worse one that puts our model at a disadvantage). We find that the teacher model has a trivial impact on final results, as it serves to stabilize the initial training phase. However, omitting the teacher model leads to suboptimal performance.
>
> |Model|AbsRel⬇️|RMSE⬇️|δ1⬆️|
> |-|-|-|-|
> |monodepth2 as teacher|0.090|3.955|0.918|
> |w/o teacher|0.093|4.432|0.910|
>
> `Table J`

---

> > ### Comment · Area_Chair_nWUG · 2025-08-06
> >
> > Dear Reviewer,
> > Could you please read the rebuttal, update your final review, and share any remaining follow-up questions, if any? Also, kindly acknowledge once this is done.
> > Thank you.

---

> ### Author Response · Authors · 2025-08-07
> **Authors' Kind Reminder to Reviewer s3Km**
>
> Dear Reviewer s3Km,
>
> Thank you very much for taking the time to read our rebuttal and for your acknowledgement. We greatly appreciate your detailed review and valuable feedback.
>
> We hope that our response has effectively addressed your previous concerns regarding the novelty of our work, the ablation studies for key modules ($L_{GDS}$ and $L_{SKY}$), the complexity of the training procedure, and the computational overhead.
>
> We sincerely look forward to your further feedback. If you feel that our rebuttal has sufficiently resolved your concerns, we would kindly ask you to consider re-evaluating our paper's rating in your final assessment.
>
> Of course, we highly value any opportunities to address any potential remaining concerns before the discussion closes, which might be helpful for improving the rating of this submission. Please do not hesitate to comment on any further concerns. Your feedback is extremely valuable!
>
> Thank you once again for your time and professional engagement.
>
> Sincerely,
>
> The Authors of Paper #14390

---

### Official Review · Reviewer_fCDa · 2025-07-01

**Clarity:** 3
**Significance:** 3
**Originality:** 3
**Rating:** 5
**Confidence:** 4

**Summary:**

This paper presents a novel approach to utilize Stable Diffusion (SD)'s visual prior to significantly enhance the accuracy and generalizability of self-supervised monocular depth estimation (SSMDE). Leveraging SD priors in SSMDE faces challenges due to artifacts from reprojection loss and misalignment between Scale-Shift-Invariant (SSI) depth and Scale-Invariant (SI) depth. Experiments on the KITTI dataset achieves best performance compared to other SSMDE methods. Zero-shot evaluation on StereoDriving and Cityscapes datasets shows remarkable generalization ability also in out-of-distribution diverse weather conditions.

**Questions:**

Q01. SD-based comparison. For fair comparison with the SD-based methods, it is recommended to conduct the training and evaluation following their protocols, e.g. in LOTUS the KITTI images are cropped to 1216 x 352.

Q02. Indoor evaluation. It is interesting to see if the proposed method can also work for indoor SSMDE tasks, which is missing in the paper.

Q03. Evaluation on (more) reliable GT. Extra evaluation on the KITTI improved ground truth is needed for more reliable comparison.

**Ethical Concerns:**

["NO or VERY MINOR ethics concerns only"]

**Final Justification:**

The paper provides interesting insights for the use of SD priors in a self-supervised domain backed up by a through motivation and clear experiments. The zero-shot performance on NYUv2 (W02/Q02) is impressive and the reviewer is very happy for adding these to the content of the paper / supplementary material. Although an additional quantitative result would provide additional insights for the SD prior perturbation, the reviewer thinks it is not urgently necessary for presentation. The paper contributions (in particular S01, S02, S03) justify a publication at NeurIPS. The reviewer therefore recommends acceptance.

**Limitations:**

Yes.

**Paper Formatting Concerns:**

None.

**Quality:**

4

**Strengths And Weaknesses:**

Strengths:

S01. SD priors for SSMDE. The well-written paper introduces a novel approach to harness the SD visual prior for SSMDE by addressing the challenges of preserving the SD latent space and aligning SSI and SI depth, which effectively enhances the accuracy and sharpness of SSMDE results and meaningfully propagates the usage of SD priors in the self-supervised framework.

S02. Motivation and justification. The approach in anchoring the SD priors is intuitively appealing and well-justified by in-depth analysis and visualizations, demonstrating the effectiveness of the proposed surrogate task and stability to mixture ratio and different size of the auxiliary dataset. The motivation and design of the SSG are well-explained with solid derivations and straightforward visualizations, and the method is shown to effectively tackle the misalignment issue and integrate the SD outputs to SSMDE framework.

S03. Strong experimental results. Strong qualitative and quantitative results in evaluations on KITTI dataset and zero-shot evaluations, showing the effectiveness of the proposed method in improving depth estimation accuracy and sharpness, and generalization ability across diverse weather conditions and high-dynamic data.

Weaknesses:

W01. Resolution discrapancies in evalution. The image resolution in KITTI and zero-shot evaluation is following common practice in SSMDE. However, the evaluation resolution used by SD-based methods, namely Marigold, E2E FT, and Lotus, is different, which causes a notable performance gap between those metrics reported in this paper and the original papers. Therefore, the comparison is not entirely fair, since the depth estimation performance can be sensitive to the input image size.

W02. Missing indoor evaluation. Indoor data is often considered challenging for SSMDE due to the change of space scale and complex egomotion. With the proposed method, it would be interesting to see if the strong SD priors can also help in indoor SSMDE tasks, which is missing in the paper.

W03. GT choice on KITTI. The evaluation on the KITTI dataset is only conducted on the raw LiDAR ground truth, however, due to noisy nature of LiDAR data and preprocessing issue reported [e.g. in MonoDepth2 Github Issue 274](https://github.com/nianticlabs/monodepth2/issues/274), it is recommended to also evaluate on the KITTI improved ground truth for more reliable comparison as done in Mono-VIFI.

W04. Missing experiments on perturbation of SD priors. With the approach of protecting the SD latent space from perturbations, the SD is ought to be least influenced and still able to conduct the original task as image generation. However, experiments on perturbation of SD priors are missing, which can be interesting in order to analyze the effect of the proposed method on Stable Diffusion, e.g. by image reconstruction metrics.

---

> ### Author Rebuttal · Authors · 2025-07-31
>
> # Overview
>
> We sincerely thank you for your thorough and highly constructive feedback on our manuscript. We are greatly encouraged by your positive assessment and are pleased that you found our work to be a **"novel approach" (S01)** with **appealing motivation and well-justified analysis(S02)**, supported by **"strong experimental results" (S03)**.
>
> We especially appreciate the valuable and actionable suggestions for improvement. In our revision, we have worked to address all the weaknesses and questions raised, including:
>
> # W01, Q01
>
> **Fair comparison with SD-based methods under specific evaluation resolution.**
>
> First, the reviewer noted that there is a performance gap between the results reported in our paper and those in the original work. It is primarily **due to differences in the GT used for evaluation, NOT the input resolution.** We will address this issue in detail in the W03 analysis.
>
> Second, we would like to clarify that for SD-based **supervised** models—particularly those trained on the vKITTI dataset (cropped to $1216 \times 352$) and the Hypersim dataset (resized to $480 \times 640$), such as Marigold and Lotus—the impact of input resolution on depth estimation performance is relatively limited. In contrast, self-supervised models are relatively sensitive to resolution changes due to the strong dependence of photometric reprojection losses on image details.
>
> As illustrated in Table D, the performance of supervised models remains close across resolutions of $1024 \times 320$ and $1216 \times 352$, whereas self-supervised models experience a significant drop in accuracy at higher resolutions. (note: the $1216 \times 352$ resolution follows the Marigold, images cropped from the original ones.)
>
> | Model    | Marigold    |          | Lotus       |          |  | MonoViFi    |          | MonoViT     |          |
> | -------- | ----------- | -------- | ----------- | -------- | - | ----------- | -------- | ----------- | -------- |
> |          | AbsRel ⬇️ | δ1 ⬆️ | AbsRel ⬇️ | δ1 ⬆️ |  | AbsRel ⬇️ | δ1 ⬆️ | AbsRel ⬇️ | δ1 ⬆️ |
> | 1024x320 | 0.120       | 0.874    | 0.110       | 0.892    |  | 0.093       | 0.909    | 0.096       | 0.908    |
> | 1216x352 | 0.122       | 0.871    | 0.111       | 0.891    |  | 0.106       | 0.886    | 0.110       | 0.888    |
>
> `Table D.` Performance comparison at different resolutions. **Marigold/Lotus are SD-based supervised models, and MonoViFi/MonoViT are self-supervised models.**
>
> Despite the inherent disadvantages of self-supervised training under these conditions, we leveraged the strengths of SD priors and evaluated our model at $1216 \times 352$ (while it was trained at $1024 \times 320$). As shown in Table E, our approach still outperforms SD-based supervised models, highlighting the effectiveness of our method.
>
> | Model    | AbsRel ⬇️     | SqRel ⬇️      | RMSE ⬇️       |  RMSE_log ⬇️  | δ1 ⬆️        | δ2 ⬆️        | δ3 ⬆️        |
> | -------- | --------------- | --------------- | --------------- | :-------------: | --------------- | --------------- | --------------- |
> | Marigold | 0.122           | 0.813           | 4.404           |      0.208      | 0.871           | 0.961           | 0.982           |
> | Lotus    | 0.111           | 0.615           | 3.802           |      0.174      | 0.891           | **0.971** | 0.986           |
> | E2E-FT   | 0.131           | 0.790           | 4.338           |      0.206      | 0.855           | 0.962           | 0.982           |
> | Jasmine* | **0.107** | **0.605** | **3.751** | **0.170** | **0.897** | 0.969           | **0.986** |
>
> `Table E.` Performance comparison under the Lotus $1216 \times 352$ evaluation protocols on the KITTI benchmark.
>
> # W02, Q02
>
> **New experiments and analysis on indoor datasets.**
>
> As the **first work** to introduce SD priors into self-supervised depth estimation, our experiments were initially focused on the most established self-supervised setting—the street view dataset. The primary challenge in indoor self-supervised depth estimation arises from frequent and significant rotational motion (whereas KITTI is predominantly translational), which necessitates improvements to the pose network. However, this aspect is beyond the scope of our core contribution, as our work does not modify the pose estimation network.
>
> Importantly, we argue that **evaluating the benefit of strong SD priors for indoor SSMDE tasks is less about training on indoor data and more about assessing the generalization capability on it.** To this end, we performed a zero-shot evaluation of several self-supervised models on indoor datasets, as presented in Table C. Despite the substantial distribution shift between the training and testing data, Jasmine continues to exhibit strong performance, further underscoring the effectiveness and generalizability of our approach.
>
> | Model      | AbsRel ⬇️     | SqRel ⬇️      | RMSE ⬇️       |  RMSE_log ⬇️  | δ1 ⬆️        | δ2 ⬆️        | δ3 ⬆️        |
> | ---------- | --------------- | --------------- | --------------- | :-------------: | --------------- | --------------- | --------------- |
> | Monodepth2 | 0.447           | 1.514           | 1.845           |      0.469      | 0.407           | 0.682           | 0.841           |
> | MonoViT    | 0.249           | 0.324           | 0.896           |      0.287      | 0.607           | 0.870           | 0.961           |
> | MonoViFi   | 0.248           | 0.268           | 0.797           |      0.287      | 0.622           | 0.868           | 0.956           |
> | Jasmine    | **0.198** | **0.257** | **0.701** | **0.269** | **0.689** | **0.899** | **0.963** |
>
> `Table C.` Zero-shot evaluation on the NYUv2 dataset, test on the official test split.
>
> # W03, Q03
>
> **Evaluation and Comparison at the KITTI improved GT**
>
> Thank you for the additional reminder! As shown in Table D, we report a performance comparison between Jasmine, SD-based methods, and self-supervised models on the KITTI improved GT benchmark, and we have consistently achieved the SoTA level. The inconsistency in metrics mentioned in W01 has also been resolved here.
>
> | Model    | AbsRel ⬇️ | SqRel ⬇️ | RMSE ⬇️  | RMSE_log ⬇️ | δ1 ⬆️   | δ2 ⬆️   | δ3 ⬆️   |
> | -------- | ----------- | ---------- | ---------- | :-----------: | ---------- | ---------- | ---------- |
> | Marigold | 0.099       | -          | -          |       -       | 0.916      | 0.987      | -          |
> | E2E-FT   | 0.096       | -          | -          |       -       | 0.921      | 0.980      | -          |
> | Lotus    | 0.081       | -          | -          |       -       | 0.931      | 0.987      | -          |
> | Jasmine* | **0.064**  | **0.294** | **2.982** |  **0.097**  | **0.957** | **0.994** | **0.998** |
> | MonoViT  | 0.068       | 0.314      | 3.125      |     0.105     | 0.948      | 0.992      | 0.998      |
> | MonoViFi | 0.071       | 0.338      | 3.539      |     0.113     | 0.937      | 0.990      | 0.998      |
> | Jasmine  | **0.061**  | **0.255** | **2.765** |  **0.092**  | **0.963** | **0.995** | **0.999** |
>
> `Table F.` Quantitative comparison between Jasmine and SD-based/self-supervised methods on the KITTI dataset using improved GT.
>
> # W04
>
> **Experiment on SD prior perturbation.**
>
> The motivation for protecting SD priors stems from the observation that self-supervised training can disrupt the integrity of these priors. The supervised approaches, like Marigold, can successfully leverage synthetic data to *keep the SD latent space intact (Original sentence from Marigold paper)*. However, even models like Marigold are **NOT** directly applicable to the original image generation task, as their adaptation for depth estimation does not preserve the generative pathway required for RGB pixel synthesis. Instead, the inheritance of SD priors in Marigold is reflected through improved prediction details and enhanced generalization ability. And both of these points are embodied in and ablated in our paper, undoubtedly proving the protection of SD priors.
>
> However, we agree that a more direct evaluation of SD prior perturbation — through image reconstruction metrics — would provide additional insights, and it should be done in image generation fine-tuning. We consider this a valuable direction for future work.
>
> # End
>
> We believe these additions have significantly strengthened the paper. The above results will be included in the final manuscript.

---

> > ### Comment · Area_Chair_nWUG · 2025-08-06
> >
> > Dear Reviewer,
> > Could you please read the rebuttal, update your final review, and share any remaining follow-up questions, if any? Also, kindly acknowledge once this is done.
> > Thank you.

---

> > ### Comment · Reviewer_fCDa · 2025-08-06
> >
> > The reviewer wants to thank the authors for their detailed answers concetning most of the points raised. The reviewer thinks that W01/Q01, W03/Q03 are adequatly and convincingly addressed.
> > The zero-shot performance on NYUv2 (W02/Q02) is indeed impressive and the reviewer would highly recommend to add this insightful experiment (Table c) to the supplementary material.
> > Regarding W04: The reviewer agrees with the authors that an additional quantitative result would "provide additional insights" for the SD prior perturbation. However, even despite this being included in the final version, the paper contributions (in particular S01, S02, S03 justify a publication at NeurIPS from the reviewers perspective.

---

> > > ### Author Response · Authors · 2025-08-07
> > > **Authors' sincere gratitude to Reviewer fCDa**
> > >
> > > Dear Reviewer fCDa,
> > >
> > > Thank you so much for your extremely thorough and constructive review of our work! We sincerely appreciate the time and effort you dedicated to providing such insightful feedback.
> > >
> > > The additional experiments and analyses prompted by your comments have significantly strengthened our submission, and we are deeply grateful for your valuable suggestions. We also want to express our sincere gratitude for your positive feedback!
> > >
> > > As you requested, we will be sure to include all the additional results and corresponding discussions in the supplementary material for the final version of the paper.
> > >
> > > Once again, thank you for your responsible, thoughtful, and encouraging review!
> > >
> > > Best regards,
> > >
> > > The Authors of Submission 14390

---

### Official Review · Reviewer_SPog · 2025-07-02

**Clarity:** 1
**Significance:** 3
**Originality:** 3
**Rating:** 3
**Confidence:** 4

**Summary:**

This paper introduces Jasmine, a self-supervised monocular depth estimation framework that leverages Stable Diffusion (SD) priors to improve depth map sharpness and generalization, without requiring high-precision ground-truth supervision. The authors address the challenge of integrating SD priors in the self-supervised setting by introducing a mix-batch image reconstruction (MIR) surrogate task and a Scale-Shift GRU (SSG) module to reconcile scale-shift invariance differences between supervised and self-supervised estimates. Experimental results on KITTI and several zero-shot benchmarks demonstrate that Jasmine sets new state-of-the-art results for self-supervised methods and achieves compelling detail preservation in predictions compared to prior work.

**Questions:**

1. How exactly the SD is integrated with self-supervision paradigm? Not technically but intuitively, as L29-L34 and L119-L123 didn't provide direct motivation explanation of this point.
2. Why the introducing of task switcher can address the problem of SD prior perturbation?
2. As far as i understood, the task switcher let the SD the be trained with either depth map or color image, as the equation stated in L137 and L138. However, for the depth predicted by SD, it requires to be pretrained on z^y (as explained in Sec. 3.1) which is groundtruth depth, and that obviously make the SD becomes fully supervised.
3. Due to the figure caused huge confusion, how exactly the self-supervision compromise SD prior need clear explanation.

other problems:
1. At L48-L49, there is no experiment or citation supporting the claim.
2. At L164-L170, the first point states “synthetic images are not essential” which is totally contradict to the explanation in L155-L160.
3. The self-reprojection supervision in L33 may should be the reprojection in self-supervision

specific to the figures:
1. At L125, the Fig. 3 (a) makes me seriously doubt about how the self-supervision compromise SD prior. The explaination of the figure is useless, there are so many inconsistencies between the text and the figure.
2. As Fig. 3 (g) is better to be comprehend than Fig. 3 (a), but the axes label should also be explained in main text as the missing of immediate reference makes it so confused to understand the main content.
3. At L201, I cannot locate the corresponding DepthHead in Fig. 4 (a). And the right bottom part of this figure really confused me, what are those arrows from nowhere means? how the z^I integrate into the computation? Such confusion also appears when the author says in L207 “subsequently update D_k to D_k+1.

**Ethical Concerns:**

["NO or VERY MINOR ethics concerns only"]

**Final Justification:**

The repeated existence of unclarity greatly hindered the quality of this submission which is critical for a real strong submission especially for NIPS-like top-tier venues. Thus, I prefer to keep the initial rating.

**Limitations:**

yes

**Quality:**

2

**Strengths And Weaknesses:**

Strengths:
    - Technical Soundness and Contributions: The paper addresses a long-standing challenge in self-supervised depth estimation—loss of fine detail and generalization—by creatively leveraging SD priors without ground-truth depth.
    - Comprehensive Results: Extensive experiments are presented, including on the KITTI dataset and challenging zero-shot settings. Jasmine achieves state-of-the-art results among SSMDEs, with significant improvements reported over existing methods. The detailed ablation studies carefully disentangle contributions of key components (SD prior, MIR, SSG), clearly showcasing their necessity and complementary effects.

Weaknesses:
    - Poor Writing Logic & Figure Presentation: The writing logic makes me very confused, which greatly influence my judgment over the following technical motivation. Same to the writing, the figures (especially for Fig. 3 (a) (g) and Fig. 4 (a)) are not self-exlained and makes the comprehension harder.
    - Unclear Technical Motivation: Through out the presentation, how the self-supervision are integrated into SD is unclear, even though I can make deduction by myself, but it should be very clear for a good submission.
    - Confused Technical Implementation: There are concerns about the implementation of the techniques as stated in the Questions.

---

> ### Author Rebuttal · Authors · 2025-07-30
>
> # Overview
>
> We sincerely thank you for your time and for providing this critical feedback. We apologize that some of the preliminaries and figure presentation were unclear. Despite these major issues with clarity, we are encouraged that the reviewer still recognized that **our paper addresses a long-standing challenge in this area and shows strong experimental results.** Following the reviewer's guidance, we try to address every concern as follows.
>
> # Weaknesses: Writing Logic & Figure Presentation
>
> In the paper, we first present our motivation: SSMDE suffers from poor generalization and limited prediction detail, whereas SD priors offer a promising solution. However, when introducing SD priors, we encountered 2 challenges: (1) self-supervised signals can disrupt SD priors, and (2) there are distributional differences between self-supervised depth and SD outputs (SI-SSI). We addressed them with MIR and SSG, ultimately enabling the effective integration of SD priors, which led to significant improvements in performance, generalization, and prediction detail. In the method section, after introducing the preliminaries on traditional SSMDE and SD protocol, we provide a detailed explanation of MIR and SSG. For MIR, we first discuss the challenges (1) principle, then describe the implementation details, and further expand MIR with diverse auxiliary images usage. For SSG, we also begin with the challenges (2) principle,  then elaborate on the design details. We hope this clarifies the overall writing logical of our paper. Questions regarding the figures will be addressed in 1️⃣,2️⃣,3️⃣ below.
>
> # How self-supervision introduce to SD
>
> > **Weaknesses: Unclear Technical Motivation
> > Questions 1.
> > Questions 3.**
>
> First, it is important to clarify that the self-supervision used here is distinct from the unsupervised learning of foundational vision models like DINOv2. In our context, self-supervision means the training process leverages the temporal and spatial relationships between sequential images to train the depth estimator, without GT depth.
>
> Intuitively, integrating SD with self-supervision fundamentally changes the source of supervision for the SD output. In the standard single-step denoising paradigm, the original SD model directly predicts depth, and during training, GT depth is used to supervise the U-Net’s output via an MSE loss. In our approach, however, the supervisory signal for the SD output comes from a reprojection loss. Specifically, we predict the relative pose between adjacent frames and use **SD output + pose** to reproject the adjacent frames onto the current input image. If the SD-predicted depth is accurate, the reprojected/reconstructed image should closely match the original input image. This photometric consistency serves as the self-supervised signal to guide depth learning.
>
> For **Question 3**, in supervised SD-based depth estimation (e.g., Marigold), $y$ indeed represents the GT depth map. However, in our self-supervised framework, $y$ should be interpreted as **the target output that the model aims to generate, rather than a pre-existing supervised signal.** Specifically, as described in Sec 3.1, in our single-step denoising strategy, the U-Net $f_\theta^z$ takes pure noise $\mathbf{n}$ and an image latent variable $z^I$ as input, and the model's task is to learn a mapping that transforms $\mathbf{n}$ into a meaningful depth map latent $z_0^y$. Thus, **the model never sees and never requires the 'correct answer' $z^y$**.
>
> # How self-supervision compromise SD prior
> > **Questions 4.
> 1️⃣Specific to the figures 3 (a)**
>
> For clarity, we first revisit the core definition of disparity: when capturing 2 images of the same scene from different camera positions, the **same point** will appear at different pixel coordinates in each image. This difference is known as *disparity*. Disparity and depth are inversely proportional and correspond one-to-one.
>
> From this perspective, the loss function in Eq. 2 is designed to find the most **similar point** in the source view $I_{t'}$ for each point in the target view $I_t$. From the coordinate difference between these points $I_{t'\rightarrow t}$, we obtain the depth supervision (Eq.1). However, if the corresponding point in $I_t$ is occluded in $I_{t'}$, the optimization process is forced to select the "least bad" alternative, resulting in an incorrect match and, consequently, erroneous depth estimation.
>
> Taking **Fig 3(a)** as an example, the scene consists of an infinite background (depth=$\infty$), an orange rectangle (20m), and a light blue tree (10m). Here, the camera only shifts horizontally between the target and source views. For point q (tree), the correct disparity is 10 pixels (10m). For point p,r (rectangle), the correct disparity should be 5 pixels (20m).
>
> Due to camera movement, the tree in the source view occludes the correct matching point of p. To minimize photometric loss, the algorithm searches for the most similar region nearby and once again finds the orange area at p', which is 10 pixels (10m).
>
> This error causes the expected depth edge on the right side of point p to disappear (the right side's depth is all 10m, same reason as p), resulting in a blurred boundary in the depth map. When such a depth map is used as a supervisory signal to guide the SD model, it effectively introduces "noisy" data. This forces the SD model to learn and reproduce these incorrect and blurred boundaries, thereby undermining its strong prior knowledge of clear object boundaries.
>
> # Others
>
> > **2️⃣Specific to Fig 4(a)**
>
> Since we cannot include the revised figure in this rebuttal, we will provide a detailed textual explanation of the process below.
>
> **For DepthHead** We want to clarify that the relationship of $(D_k,h^k)$ is same to $(y,z^y)$  in the SD model. The SSG network does not directly process the original image, but the encoded features. The difference is that SD uses VAE for encoding and decoding, while GRU only uses one convolutional layer, so we ignored this structure in the figure. We thank you for pointing this out and will include it in the final version.
>
> **Arrows from nowhere** In fact, Fig 4(a) corresponds to the gray rectangle shown in Fig. 2, standing for an iteration within two consecutive ones. Thus, the origins and destinations of 'nowhere' arrows indicate connections to the preceding and following SSG iterations, respectively.
>
> **$z^I$ integration and $D_k$ update**
> To better clarify these questions and Fig. 4(a), we detail the first iteration (k=0) below:
>
> 1. We first encode the SD output $D_0(D_{SSI})$ to obtain the hidden state $h^0$.
> 2. We use learnable $Q_{SC}/Q_{SH}$(as query) to interact with the $h^0$ (as key/value) with cross-attention and generate scale $s_c$ and shift $s_h$.
> 3. The input $x^0$ is formed by concatenating the image features $z^I$ and the updated depth $s_c*D_0+s_h$.
> 4. The GRU takes $h^0$ and $x^0$ as input and outputs next hidden state $h^1$. After decoding, depth residual $D_{\delta}$ is obtained. Finally, we use Eq 10 to calculate the refined depth map $D_1$.
>
> > **3️⃣Specific to Fig 3(g)**
>
> We agree that Figure 3(g) should be more self-contained, and we will revise its caption in the final version, like:
>
> In sub-figure (g), the x-axis ($\lambda$) is the proportion of the KITTI dataset used in training(e.g., 0.3 indicates a KITTI: Hypersim ratio of 3:7 in the MIR training data), the y-axes are the metrics of the performance, including AbsRel (Absolute Relative Error, lower is better) and A1 (Accuracy, higher is better), definitions detailed in Sec E. We show the performance variations under different $\lambda$ settings for photometric supervision (Eq. 5, better) and latent supervision (Eq. 4, worse).
>
> > **Questions 2.**
>
> The task switcher itself cannot solve the SD prior perturbation problem, but the surrogate task (MIR) enables can. As we explained in "How self-supervision compromises SD priors," self-supervision introduces "noisy" supervision signals. The role of the switcher is to introduce "clean" ones. Specifically, the training target of the image reconstruction task is the input image itself, which contains the most perfect, sharpest, and structurally correct visual details. Therefore, it guides the model to focus on and preserve realistic edges, textures, and structures. During training, the "bad gradients" from the self-supervised loss and the "good gradients" from the image reconstruction task are mixed together. These "good gradients" act as a normalization or an anchor, constantly pulling the model's parameters back to a "correct" region capable of generating sharp, realistic details, effectively counteracting the negative impact of the "bad gradients" on the model's priors.
>
> > Other Problems 1.
>
> This claim originates from the foundation work on self-supervised depth estimation by Godard et al. (2017), titled *"Unsupervised Monocular Depth Estimation with Left-Right Consistency"* (CVPR 2017).
>
> > other problems 2.
>
> At L155-L160, the paper states that we utilize the synthetic Hypersim dataset as auxiliary image. However, using synthetic data does **NOT** mean they must be used. As demonstrated in our experiments (Table 3, (d), (13), L325-l327), using high-quality real images can achieve similarly effective results. So they are not contradictory.
>
> > other problems 3.
>
> We will fix it in the final version.

---

> > ### Comment · Area_Chair_nWUG · 2025-08-06
> >
> > Dear Reviewer,
> > Could you please read the rebuttal, update your final review, and share any remaining follow-up questions, if any? Also, kindly acknowledge once this is done.
> > Thank you.

---

> > ### Comment · Reviewer_SPog · 2025-08-07
> >
> > I appreciate the detailed rebuttal from the authors.
> >
> > Some of my confusions are now cleared, but it is evident that the clarity of the manuscript could be greatly improved. For example, the response to "How does self-supervision compromise the SD prior?" effectively addressed my confusion, but the original explanation of Fig. 3(a) in Lines 142–154 is inadequate and clearly needs improvement. Similarly, the additional explanation of "Why does introducing the task switcher help address the problem of SD prior perturbation?" makes the design intuition clear — an explanation that is entirely missing in the current submission. It would be beneficial for the authors to revise the manuscript to ensure that the writing is fully self-contained by integrating these additional insights.
> >
> > However, the answer to "How is self-supervision introduced to SD?" does not contribute to a better understanding. From my perspective, I recognize that integrating self-supervision and SD is a significant contribution, but Lines 29–34 and 119–123 merely state that efficient SD sampling facilitates the integration, which does not explain how the integration is actually carried out. Moreover, the explanation in the rebuttal elaborates on the connection between Eq. (1) and Eq. (3) further makes the situation unclear. As a more straightforward response, the authors could simply provide the input–output workflow and the involved computations, especially considering the poor clarity of Fig. 4(a).
> >
> > Best Regards

---

> > > ### Author Response · Authors · 2025-08-07
> > >
> > > Dear Reviewer SPog,
> > >
> > > Thank you very much for your thoughtful follow-up and acknowledging that our previous rebuttal has cleared some of your confusion. We sincerely appreciate your constructive feedback.
> > >
> > > Specifically, we will address the points you raised as follows:
> > >
> > > # Clarifying the Integration of Self-Supervision and Stable Diffusion
> > >
> > > Following your excellent advice, we will revise the explanation to present a clear **input-output workflow** with the involved computations.
> > >
> > > For clarity, we have temporarily ignored the MIR and SSG parts here:
> > >
> > > * **Input:** A sequence of temporally adjacent images (like video frames), e.g., a source image $I_{t'}$ and a target image $I_t$.
> > >
> > >   - Here $I_t$ is exactly the training image, $I_{t'}$ can be the previous frame $I_{t-1}$ or the next frame $I_{t+1}$ of $I_t$.
> > > * **Step 1: Depth Prediction (via SD U-Net):**
> > >
> > >   * The SD U-Net, $f_\theta^z$, takes the latent of the target image, $z^I=\text{VAE.encode}(I_t)$, and a pure noise vector, $\mathbf{n}$, as input.
> > >   * It performs a single-step denoising process to predict the latent representation of a depth map, $z_0^y$.
> > >     * This is the key point we mentioned in Lines 29-34 and 119-123: single-step denoising makes this step become a fast, feed-forward process rather than computationally prohibitive with slow, iterative denoising.
> > >   * The VAE decoder then converts $z_0^y$ into the final depth map, $D = \text{VAE.decode}(z_0^y)$.
> > >   * **Crucially, no GT depth is ever used in this step.**
> > > * **Step 2: Pose Prediction:**
> > >
> > >   * A separate PoseNet takes the image pair ($I_t$, $I_{t'}$) as input and predicts the relative camera pose (rotation and translation), $T_{t \to t'}$.
> > > * **Step 3: Self-Supervised Signal Generation (Image Reprojection):**
> > >
> > >   * Using the predicted depth map $D$ and the predicted pose $T_{t \to t'}$, we perform a warping operation to reproject the pixels from the source image $I_{t'}$ onto the target image's coordinate system. This creates a synthesized target image, $\hat{I}_t$.
> > > * **Step 4: Loss Calculation and Optimization:**
> > >
> > >   * A photometric reprojection loss, $L_{ph}$, is calculated by comparing the synthesized image $\hat{I}_t$ with the original target image $I_t$.
> > >   * **This loss, $L_{ph}$, is the core self-supervised signal.** If the predicted depth $D$ is incorrect, the reprojected image $\hat{I}_t$ will not match the original $I_t$, resulting in a high loss.
> > > * **Step 5: End-to-End Backpropagation:**
> > >
> > >   * The gradient from $L_{ph}$ is backpropagated through the entire computational graph. This means the gradient flows back to update the weights of both the **PoseNet** and, most importantly, the **SD U-Net** ($f_\theta^z$).
> > > * **Output:** The predicted depth map $D$ obtained from Step 1.
> > >
> > > # Improving Manuscript Clarity and Integrating Rebuttal Insights
> > >
> > > We will thoroughly revise the paper by integrating the key insights from this rebuttal:
> > > 1.  We will replace the inadequate explanation of **occlusion for Fig. 3(a)** (Lines 142–154) with our clearer step-by-step description.
> > >
> > > 2. We will add the currently missing **design intuition for the task switcher**, explaining precisely how it enables the MIR task to preserve SD priors.
> > >
> > > 3. The **integration of self-supervision and stable-diffusion** will also be added at paper's Lines 29–34 after you think it is clear.
> > >
> > > Thank you once again for your detailed and highly constructive feedback. Your guidance has given us a clear path to significantly improve the quality and clarity of our paper. Of course, please do not hesitate to comment on any further concerns. Your feedback is extremely valuable!
> > >
> > > Best Regards,
> > >
> > > The Authors of Paper #14390

---

> > > > ### Comment · Reviewer_SPog · 2025-08-08
> > > >
> > > > Great, thank you for the discussion.
> > > >
> > > > My concerns have been addressed. I will carefully check all the provided explanation again then accordingly decide my final recommendation.
> > > >
> > > >
> > > > Best Regards

---

> ### Author Response · Authors · 2025-08-08
> **Authors' sincere gratitude to Reviewer SPog**
>
> Dear Reviewer SPog,
>
> Thank you for your prompt reply and for confirming that our explanations have addressed your concerns. We truly appreciate the time and effort you have dedicated throughout this entire review process. Your insightful questions and constructive feedback have been invaluable in helping us strengthen the paper.
>
> We look forward to your final recommendation after your careful re-evaluation. Should any further questions arise during this time, we would welcome the opportunity to clarify them.
>
> Thank you once again for the very productive discussion.
>
> Best Regards,
>
> The Authors of Paper #14390

---

### Official Review · Reviewer_arKR · 2025-07-03

**Clarity:** 3
**Significance:** 3
**Originality:** 3
**Rating:** 5
**Confidence:** 3

**Summary:**

This paper proposes the first self-supervised monocular depth estimation framework based on Stable Diffusion. It introduces Mix-batch Image Reconstruction to protect Stable Diffusion's visual priors by supervising the model with image reconstruction instead of noisy reprojection losses. Additionally, it designs a Scale-Shift GRU (SSG) to align SD’s scale-shift-invariant depth outputs with the scale-invariant requirements of self-supervised training. It achieves state-of-the-art results on KITTI and demonstrates strong generalization across diverse zero-shot scenarios.

**Questions:**

While the method performs well on KITTI and shows some cross-domain generalization, could the authors further evaluate its robustness on more diverse domains, such as indoor scenes?

**Ethical Concerns:**

["NO or VERY MINOR ethics concerns only"]

**Final Justification:**

Most of my concerns, including fair comparison at the same resolution, inference latency, and indoor performance, have been satisfactorily addressed, and I have decided to maintain my original score.

**Limitations:**

Yes

**Quality:**

3

**Strengths And Weaknesses:**

Strengths

Leveraging the rich visual priors encoded in SD, such as semantic understanding, structural perception, and fine-grained texture modeling, is a natural and promising direction for enhancing monocular depth estimation. However, directly applying SD to self-supervised depth frameworks introduces significant challenges. This paper successfully addresses these issues by introducing MIR and SSG, making it the first to effectively integrate diffusion priors into self-supervised depth estimation. The approach establishes a new paradigm for the task, supported by thorough experiments, well-designed ablations, and strong performance in both quantitative metrics and qualitative results. Overall, I find the contributions meaningful and the methodology sound, and I would lean toward an Accept rating.

Weaknesses
1) Some baseline results in Table 1 are reported at 640×192 resolution, while Jasmine uses 1024×320 for both training and evaluation. This discrepancy is not clearly annotated in the table, which may mislead readers when comparing numbers directly.

2) Including a 640×192 variant would facilitate direct comparison with prior work and help future research adopt and benchmark Jasmine more easily, especially in low-resource or real-time settings.

3) The paper does not report inference latency or runtime performance, which is important considering that Jasmine relies on SD components and a recurrent GRU module.

---

> ### Author Rebuttal · Authors · 2025-07-30
>
> # Overall
>
> We sincerely thank you for your time and effort in providing such a constructive and detailed review. We are greatly encouraged that you found our **contributions meaningful**, our **methodology sound**, and that our work **establishes a new paradigm for self-supervised depth estimation.**
>
> Your suggestions regarding the fairness of benchmark comparisons, the inclusion of runtime analysis, and further evaluation of generalization on more diverse domains like indoor scenes are highly insightful. We fully agree that addressing these points will significantly strengthen our paper's clarity and completeness. Below, we provide a point-by-point response to your comments and will incorporate all the valuable changes into our final manuscript.
>
> # On Fairness of Benchmarking and Additional Experiments (W1, W2)
>
> We have carefully checked Table 1 and ensured that all of the baseline results are reported at the same resolution of 1024x320.
>
> We have included the performance evaluation of our retrained model at a resolution of 640 × 192 in Table A, alongside comparisons with other latest self-supervised models. The results continue to demonstrate the SoTA performance of our method.
>
> | Model             | Venue     | AbsRel ⬇️     | SqRel ⬇️      | RMSE ⬇️       |  RMSE_log ⬇️  | δ1 ⬆️        | δ2 ⬆️        | δ3 ⬆️        |
> | ----------------- | --------- | --------------- | --------------- | --------------- | :-------------: | --------------- | --------------- | --------------- |
> | Jaeho et al       | CVPR 2024 | 0.096           | 0.696           | 4.327           |      0.174      | 0.904           | 0.968           | 0.985           |
> | RPrDepth          | ECCV 2024 | 0.097           | 0.658           | 4.279           |      0.169      | 0.900           | 0.967           | **0.985** |
> | MonoViFi          | ECCV 2024 | 0.096           | 0.627           | **4.179** |      0.170      | 0.903           | **0.969** | **0.985** |
> | **Jasmine** | --        | **0.092** | **0.622** | 4.180           | **0.168** | **0.908** | **0.969** | **0.985** |
>
> `Table A.` Quantitative results on the KITTI dataset with 640x192 resolution. The best results are marked in **bold**.
>
> # On Inference Latency and Runtime Performance (W3)
>
> We present the inference latency and computational complexity of the Jasmine model in Table B, alongside comparisons with both SD-based and self-supervised methods. Our results indicate that the inclusion of SSG has minimal impact on inference delay, as Jasmine and Lotus exhibit comparable inference time. Although incorporating SD does lead to a notable increase in computational complexity relative to other self-supervised approaches, Jasmine strikes a trade-off between performance and computational cost.
>
> |   Method:   | Marigold | Lotus | E2E-Mono | Monodepth2 | MonoViT | MonoViFi | Jasmine |
> | :----------: | :------: | :---: | :------: | :--------: | :-----: | :------: | :-----: |
> |  MACs ⬇️  |   133T   | 2.65T |  2.65T  |   21.43G   | 25.63G |  28.79G  |  2.83T  |
> | Runtime ⬇️ |  9.88s  | 157ms |  152ms  |    33ms    |  29ms  |   25ms   |  172ms  |
>
> `Table B.` Computational efficiency comparison. MACs and Runtime are measured under a single image with 1024x320 resolution on one RTX 4090 card. MACs is the number of core multiply-accumulate operations and MACs≈FLOPs/2.
>
> # On Generalization to More Diverse Domains (Q1)
>
> As the **first work** to introduce SD priors into self-supervised depth estimation, in the paper, our experiments were initially focused on the most established self-supervised setting—the street view dataset. But to address the reviewer’s concerns, we performed a zero-shot evaluation of several **self-supervised** models on indoor datasets, as presented in Table C. Despite the substantial distribution shift between the training and testing data, Jasmine continues to exhibit strong performance, further underscoring the effectiveness and generalizability of our approach.
>
> | Model      | AbsRel ⬇️     | SqRel ⬇️      | RMSE ⬇️       |  RMSE_log ⬇️  | δ1 ⬆️        | δ2 ⬆️        | δ3 ⬆️        |
> | ---------- | --------------- | --------------- | --------------- | :-------------: | --------------- | --------------- | --------------- |
> | Monodepth2 | 0.447           | 1.514           | 1.845           |      0.469      | 0.407           | 0.682           | 0.841           |
> | MonoViT    | 0.249           | 0.324           | 0.896           |      0.287      | 0.607           | 0.870           | 0.961           |
> | MonoViFi   | 0.248           | 0.268           | 0.797           |      0.287      | 0.622           | 0.868           | 0.956           |
> | Jasmine    | **0.198** | **0.257** | **0.701** | **0.269** | **0.689** | **0.899** | **0.963** |
>
> `Table C.` Zero-shot evaluation on the NYUv2 dataset, test on the official test split.

---

> > ### Comment · Reviewer_arKR · 2025-08-05
> >
> > Thank you to the authors for the detailed rebuttal. Most of my concerns have been satisfactorily addressed, and I have decided to maintain my original score.

---

> > > ### Author Response · Authors · 2025-08-07
> > > **Authors' Sincere Thanks to Reviewer arKR!**
> > >
> > > Dear Reviewer arKR,
> > >
> > > We are very grateful for your thoughtful comments and for your positive assessment of our work throughout the review process.
> > >
> > > Thank you for taking the time to review our rebuttal. We are pleased to hear that it satisfactorily addressed your concerns. We sincerely appreciate your continued support for our paper and for maintaining your positive score. Your perspective has been invaluable in helping us refine our manuscript.
> > >
> > > Thank you once again for your constructive engagement.
> > >
> > > Best regards,
> > >
> > > The Authors of Submission 14390

---

### Decision · Program_Chairs · 2025-09-17

**Decision:**

Accept (poster)

**Comment:**

This paper received mixed but mostly positive reviews (1 borderline reject, 1 borderline accept, 2 accepts).

The paper addresses one of the primary weaknesses of self-supervised depth estimation through the use of the generative prior and adds a method to preserve that prior (cross-task training or more simply co-training) as well as introducing a new architectural element that addresses the issue of scale and shift prediction. The reviewers praised the task as well as results, with the only reservation being the presentation of the paper.

After reading the paper, reviews, rebuttals, and discussions, I advocate for acceptance. I believe this is a very good paper that addresses an interesting task. The only thing that is preventing me from recommending a higher status than acceptance is that it is not clear how important the task of self-supervised depth is anymore due to the emergence of geometric foundation models like Dust3r and VGGT. While it's not this paper's fault that they do not address this, that particular lack of comparison and understanding prevents me from recommending something higher than acceptance.